# A Manifold Perspective on the Statistical Generalization of Graph Neural Networks

Zhiyang Wang [* 1]   Juan Cerviño [* 2]   Alejandro Ribeiro [1]

## Abstract

Graph Neural Networks (GNNs) extend convolutional neural networks to operate on graphs. Despite their impressive performances in various graph learning tasks, the theoretical understanding of their generalization capability is still lacking. Previous GNN generalization bounds ignore the underlying graph structures, often leading to bounds that increase with the number of nodes – a behavior contrary to the one experienced in practice. In this paper, we take a manifold perspective to establish the statistical generalization theory of GNNs on graphs sampled from a manifold in the spectral domain. As demonstrated empirically, we prove that the generalization bounds of GNNs decrease linearly with the size of the graphs in the logarithmic scale, and increase linearly with the spectral continuity constants of the filter functions. Notably, our theory explains both node-level and graph-level tasks. Our result has two implications: i) guaranteeing the generalization of GNNs to unseen data over manifolds; ii) providing insights into the practical design of GNNs, i.e., restrictions on the discriminability of GNNs are necessary to obtain a better generalization performance. We demonstrate our generalization bounds of GNNs using synthetic and multiple real-world datasets.

## 1. Introduction

Graph convolutional neural networks (GNNs) (Scarselli et al., 2008; Defferrard et al., 2016; Bruna et al., 2014) have emerged as one of the leading tools for processing graph-structured data. There is abundant evidence of their empirical success across various fields, including but not limited to weather prediction (Lam et al., 2023), protein structure prediction in biochemistry (Jumper et al., 2021; Strokach et al., 2020), resource allocation in wireless communications (Wang et al., 2022a), social network analysis in sociology (Fan et al., 2020), point cloud in 3D model reconstruction (Shi & Rajkumar, 2020) and learning simulators (Fortunato et al., 2022).

The effectiveness of GNNs relies on their empirical ability to *predict* over unseen data. This capability is evaluated theoretically with *statistical generalization* in deep learning theory (Kawaguchi et al., 2017), which quantifies the difference between the *empirical risk* (i.e. training error) and the *statistical risk* (i.e. testing error). Despite the abundant evidence of GNNs' generalization capabilities in practice, developing concrete theories to explain their generalization is an active area of research. Many recent works have studied the generalization bounds of GNNs without any dependence on the underlying model responsible for generating the graph data (Scarselli et al., 2018; Garg et al., 2020; Verma & Zhang, 2019). Generalization analysis on graph classification, when graphs are drawn from random limit models, is also studied in a series of works (Ruiz et al., 2023; Maskey et al., 2022; 2024; Levie, 2024). In this work, we take the manifold perspective to formulate graph data on continuous topological spaces, i.e., manifolds. We emphasize that manifolds are realistic models to generate graph data that enable rigorous theoretical analysis and a deep understanding of the behaviors of GNNs.

We explore the generalization bound of GNNs through the lens of manifold theory on both node-level and graph-level tasks in the spectral domain. The graphs are constructed based on points randomly sampled from underlying manifolds, indicating that the manifold can be viewed as a statistical model for these discretely sampled points. As deep learning architectures have been established over manifolds (Wang et al., 2022b; Chew et al., 2024), the convergence of GNNs to manifold neural networks (MNNs) and the algebraical equivalence of these two frameworks facilitate a detailed generalization understanding of GNNs through spectral analysis. We demonstrate that, with an appropriate graph construction based on the sampled points from the

---

[*]Equal contribution  [1]Department of Electrical and Systems Engineering, University of Pennsylvania, Philadelphia, USA [2]Laboratory of Information and Decision Systems (LIDS), Massachusetts Institute of Technology, Cambridge, USA. Correspondence to: Zhiyang Wang <zhiyangw@seas.upenn.edu>, Juan Cerviño <jcervino@mit.edu>.

*Proceedings of the 42nd International Conference on Machine Learning*, Vancouver, Canada. PMLR 267, 2025. Copyright 2025 by the author(s).

manifold, the generalization gap between empirical and statistical risks decreases with the number of sampled points in the graphs (Figure 1c) on both node-level and graph-level tasks. More importantly, the generalization gap increases linearly with the continuity constants of frequency response functions of graph filters composing the GNN (Figure 1d). We observe that with spectral continuous filters, the GNNs are generalizable across different nodes or graphs generated from the same underlying manifold. This provides insight into the practical graph filter design from a spectral perspective. Moreover, the theoretical results indicate a trade-off between the discriminability and generalization capability of GNNs, suggesting that restrictions on the discriminability of GNNs are necessary to maintain generalization performance.

We introduce a novel unified analysis of the generalization of GNNs to unseen nodes and graphs, by relating the GNNs with MNNs in the spectral domain. We further propose restrictions on the discriminability of GNNs from the spectral perspective which results from assumptions on the continuity of the filter frequency response functions. We provide extensive experiments both on synthetic and real-world datasets to verify our generalization conclusions. Our contribution is four-fold:

1. We prove the generalization bound of GNNs on graphs generated from an underlying manifold on both node-level (Theorem 1) and graph-level (Theorem 2) by relating the algebraically equivalent GNNs and MNN in the spectral domain.

2. We provide novel generalization gap bounds that decrease linearly with the nodes of the graph in the logarithmic scale, and increase linearly with the spectral continuity constants (Assumption 1) of the filter functions.

3. We uncover an important trade-off between the discriminability and the generalization gap of GNNs, which guides practical GNN designs.

4. We verify the dependence of our generalization gaps on parameters, especially the continuity parameter, with a synthetic dataset – chair manifold – and eight real-world datasets – ArXiv, Citeseer, etc.

## 2. Related works

### 2.1. Generalization bounds of GNNs

**Node level tasks**   We first give a brief recap of the generalization bounds of GNNs on node level tasks. In (Scarselli et al., 2018), the authors give a generalization bound of GNNs with a Vapnik–Chervonenkis dimension of GNNs.

The authors in (Verma & Zhang, 2019) analyze the generalization of a single-layer GNN based on stability analysis, which is further extended to a multi-layer GNN in (Zhou & Wang, 2021). In (Ma et al., 2021), the authors give a novel PAC-Bayesian analysis on the generalization bound of GNNs across arbitrary subgroups of training and testing datasets. The authors derive generalization bounds for GNNs via transductive uniform stability and transductive Rademacher complexity in (Esser et al., 2021; Cong et al., 2021; Tang & Liu, 2023). The authors in (Yehudai et al., 2021) propose a size generalization analysis of GNNs correlated to the discrepancy between local distributions of graphs. Different from these works, we consider a continuous manifold model when generating the graph data, which is theoretically powerful and realistic when characterizing real-world data. Furthermore, the generalization bounds proved in these works either grow with the size of the graph (Esser et al., 2021; Tang & Liu, 2023; Scarselli et al., 2018), with the node degree of the graphs (Cong et al., 2021) or the maximum eigenvalues of the graph (Verma & Zhang, 2019). Notably, our generalization bound decreases with the size of the graph given that it depends on the spectral properties of the filter functions over the manifold.

**Graph level tasks**   There are also related works on the generalization analysis of GNNs on graph-level tasks. In (Garg et al., 2020), the authors form the generalization bound via Rademacher complexity. The authors in (Liao et al., 2020) build a PAC-Bayes framework to analyze the generalization capabilities of graph convolutional networks (Kipf & Welling, 2016) and message-passing GNNs (Gilmer et al., 2017), based on which the authors in (Ju et al., 2023) improve the results and prove a lower bound. The bounds either grow with the number of nodes (Liao et al., 2020) or the degree of the graphs (Garg et al., 2020) while our bound decreases with the number of nodes in the graph given that it better approximates the underlying model – the manifold. The works in (Maskey et al., 2022; 2024; Levie, 2024) are most related to ours, which also consider the generalization of GNNs on a graph limit model, in their case a *graphon*. Different from our setting, the authors see the graph limit as a random continuous model. They study the generalization of graph classification problems with message-passing GNNs with graphs belonging to the same category sampled from a continuous limit model. The generalization bound grows with the model complexity and decreases with the number of nodes in the graph. We show that a GNN trained on a single graph sampled from each manifold is enough, and can generalize and classify unseen graphs sampled from the manifold set.

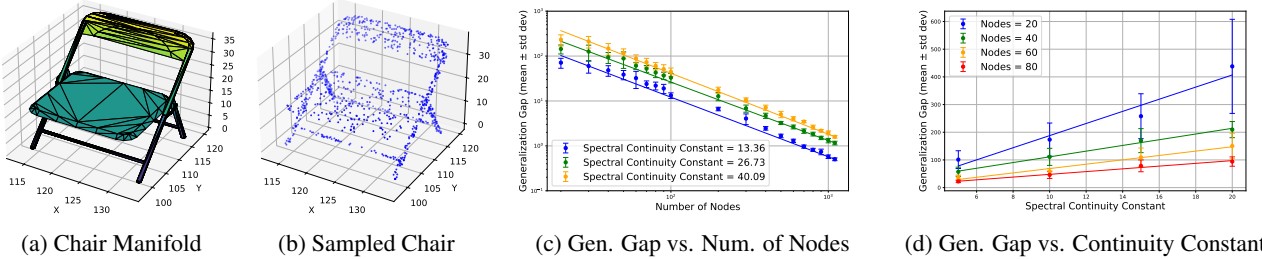

(a) Chair Manifold        (b) Sampled Chair        (c) Gen. Gap vs. Num. of Nodes        (d) Gen. Gap vs. Continuity Constants

Figure 1: Synthetic experimental results are shown on the uniformly sampled chair manifold. We construct a graph with different numbers of nodes, fix the weights of a GNN, and compute the generalization gap. We construct the graph by computing the edges for nodes that are $\epsilon$ close (cf. equation 3). In Figure 1c, we fix the spectral continuity constant (see Assumption 1) and vary the number of nodes. As our theory predicts, we see that a smaller spectral continuity constant translates into a smaller generalization gap – as the blue line is below the green line which is below the orange line. In Figure 1d we fix the number of nodes in the graph and vary the spectral continuity constant in the GNN. For the same number of nodes, a larger spectral continuity constant translates into a larger generalization gap.

## 2.2. Neural networks on manifolds

Geometric deep learning has been proposed in (Bronstein et al., 2017) with neural network architectures raised in manifold space. The authors in (Monti et al., 2017) and (Chakraborty et al., 2020) provide neural network architectures for manifold-valued data. In (Wang et al., 2024b) and (Wang et al., 2022b), the authors define convolutional operation over manifolds and see the manifold convolution as a generalization of graph convolution, which establishes the limit of neural networks on large-scale graphs as manifold neural networks (MNNs). The authors in (Wang et al., 2024a; Chew et al., 2023; Johnson et al., 2025) further establish the relationship between GNNs and MNNs with non-asymptotic convergence results for different graph constructions. Some studies have used graph samples to infer properties of the underlying manifold itself. These properties include the validity of the manifold assumption (Fefferman et al., 2016), the manifold dimension (Farahmand et al., 2007) and the complexity of these inferences (Narayanan & Niyogi, 2009; Aamari & Knop, 2021). Other research has focused on prediction and classification using manifolds and manifold data, proposing various algorithms and methods. Impressive examples include the Isomap algorithm (Choi & Choi, 2004; Wu & Chan, 2004; Yang et al., 2016a) and other manifold learning techniques (Talwalkar et al., 2008). These techniques aim to infer manifold properties without analyzing the generalization capabilities of GNNs operated on the sampled manifold.

## 3. Preliminaries

### 3.1. Graph neural networks

**Setup** An undirected graph $\mathbf{G} = (\mathcal{V}, \mathcal{E}, \mathcal{W})$ contains a node set $\mathcal{V}$ with $N$ nodes and an edge set $\mathcal{E} \subseteq \mathcal{V} \times \mathcal{V}$. The weight function $\mathcal{W} : \mathcal{E} \to \mathbb{R}$ assigns values to the edges. We define the graph Laplacian $\mathbf{L} = \text{diag}(\mathbf{A1}) - \mathbf{A}$ where $\mathbf{A} \in \mathbb{R}^{N \times N}$ is the weighted adjacency matrix. Graph signals are functions mapping nodes to a feature value. We write it as a vector $\mathbf{x} \in \mathbb{R}^N$, with each entry $[\mathbf{x}]_i$ representing the function value on node $i$.

**Graph convolutions and frequency response** A graph convolutional filter $\mathbf{h_G}$ is composed of consecutive graph shifts by graph Laplacian, defined as $\mathbf{h_G}(\mathbf{L})\mathbf{x} = \sum_{k=0}^{K-1} h_k \mathbf{L}^k \mathbf{x}$ with $\{h_k\}_{k=0}^{K-1}$ as filter parameters. We replace $\mathbf{L}$ with eigendecomposition $\mathbf{L} = \mathbf{V \Lambda V}^H$, where $\mathbf{V}$ is the eigenvector matrix and $\mathbf{\Lambda}$ is a diagonal matrix with eigenvalues $\{\lambda_{i,N}\}_{i=1}^N$ as the entries. The spectral representation of a graph filter is

$$\mathbf{V}^H \mathbf{h_G}(\mathbf{L})\mathbf{x} = \sum_{k=1}^{K-1} h_k \mathbf{\Lambda}^k \mathbf{V}^H \mathbf{x} = \hat{h}(\mathbf{\Lambda}) \mathbf{V}^H \mathbf{x}. \quad (1)$$

This leads to a point-wise frequency response of the graph convolution as $\hat{h}(\lambda) = \sum_{k=0}^{K-1} h_k \lambda^k$.

**Graph neural networks** A graph neural network (GNN) is a layered architecture, where each layer consists of a bank of graph convolutional filters followed by a point-wise nonlinearity $\sigma : \mathbb{R} \to \mathbb{R}$. Specifically, the $l$-th layer of a GNN that produces $F_l$ output features $\{\mathbf{x}_l^p\}_{p=1}^{F_l}$ with $F_{l-1}$ input features $\{\mathbf{x}_{l-1}^q\}_{q=1}^{F_{l-1}}$ is written as

$$\mathbf{x}_l^p = \sigma\left(\sum_{q=1}^{F_{l-1}} \mathbf{h_G}^{lpq}(\mathbf{L})\mathbf{x}_{l-1}^q\right), \quad (2)$$

for each layer $l = 1, 2 \cdots, L$. The graph filter $\mathbf{h_G}^{lpq}(\mathbf{L})$ maps the $q$-th feature of layer $l-1$ to the $p$-th feature of layer $l$. We denote the GNN as a mapping $\mathbf{\Phi_G}(\mathbf{H}, \mathbf{L}, \mathbf{x})$, where $\mathbf{H} \in \mathcal{H} \subset \mathbb{R}^P$ denotes a set of the graph filter

coefficients with a finite $P$ dimension at all layers and $\mathcal{H}$ denotes the set of all possible parameter sets.

## 3.2. Manifold neural networks

**Setup** We consider a $d$-dimensional compact, smooth and differentiable Riemannian submanifold $\mathcal{M}$ embedded in a M-dimensional space $\mathbb{R}^M$ with finite volume. This induces a measure $\mu$ which has a non-vanishing Lipschitz continuous density $\rho$ with respect to the Riemannian volume over the manifold with $\rho : \mathcal{M} \to (0, \infty)$, assumed to be bounded as $0 < \rho_{min} \leq \rho(x) \leq \rho_{max} < \infty$ for all $x \in \mathcal{M}$. The manifold data supported on each point $x \in \mathcal{M}$ is defined by scalar functions $f : \mathcal{M} \to \mathbb{R}$ (Wang et al., 2024b). We use $L^2(\mathcal{M})$ to denote $L^2$ functions over $\mathcal{M}$ with respect to measure $\mu$. The manifold with probability density function $\rho$ is equipped with a weighted Laplace operator (Grigor'yan, 2006), generalizing the Laplace-Beltrami operator as

$$\mathcal{L}_\rho f = -\frac{1}{2\rho}\text{div}(\rho^2 \nabla f), \qquad (3)$$

with div denoting the divergence operator of $\mathcal{M}$ and $\nabla$ denoting the gradient operator of $\mathcal{M}$ (Bronstein et al., 2017; Gross & Meinrenken, 2023).

**Manifold convolutions and frequency responses** The manifold convolution operation is defined relying on the Laplace operator $\mathcal{L}_\rho$ and on the heat diffusion process over the manifold (Wang et al., 2024b). For a function $f \in L^2(\mathcal{M})$ as the initial heat condition over $\mathcal{M}$, the heat condition diffused by a unit time step can be explicitly written as $e^{-\mathcal{L}_\rho}f$. A manifold convolutional filter (Wang et al., 2024b) can be defined in a diffuse-and-sum manner as

$$g(x) = \mathbf{h}(\mathcal{L}_\rho)f(x) = \sum_{k=0}^{K-1} h_k e^{-k\mathcal{L}_\rho}f(x), \qquad (4)$$

with the $k$-th diffusion scaled with a filter parameter $h_k \in \mathbb{R}$. We consider the case in which the Laplace operator is self-adjoint, positive-semidefinite and the manifold $\mathcal{M}$ is compact. In this case, $\mathcal{L}_\rho$ has real, positive and discrete eigenvalues $\{\lambda_i\}_{i=1}^\infty$, written as $\mathcal{L}_\rho \phi_i = \lambda_i \phi_i$ where $\phi_i$ is the eigenfunction associated with eigenvalue $\lambda_i$. The eigenvalues are ordered in increasing order as $0 = \lambda_1 \leq \lambda_2 \leq \lambda_3 \leq \ldots$, and the eigenfunctions are orthonormal and form an eigenbasis of $L^2(\mathcal{M})$. When mapping a manifold signal onto the eigenbasis $[\hat{f}]_i = \langle f, \phi_i \rangle_{\mathcal{M}} = \int_{\mathcal{M}} f(x)\phi_i(x)\mathrm{d}\mu(x)$, the manifold convolution can be seen in the spectral domain as

$$[\hat{g}]_i = \sum_{k=0}^{K-1} h_k e^{-k\lambda_i}[\hat{f}]_i. \qquad (5)$$

Hence, the frequency response of manifold filter is given by $\hat{h}(\lambda) = \sum_{k=0}^{K-1} h_k e^{-k\lambda}$.

**Manifold neural networks** A manifold neural network (MNN) is constructed by cascading $L$ layers, each of which contains a bank of manifold convolutional filters and a point-wise nonlinearity $\sigma : \mathbb{R} \to \mathbb{R}$. The output manifold function of each layer $l = 1, 2 \cdots, L$ can be explicitly denoted as

$$f_l^p(x) = \sigma \left( \sum_{q=1}^{F_{l-1}} \mathbf{h}_l^{pq}(\mathcal{L}_\rho)f_{l-1}^q(x) \right), \qquad (6)$$

where $f_{l-1}^q$, $1 \leq q \leq F_{l-1}$ is the $q$-th input feature from layer $l-1$ and $f_l^p$, $1 \leq p \leq F_l$ is the $p$-th output feature of layer $l$. We denote MNN as a mapping $\Phi(\mathbf{H}, \mathcal{L}_\rho, f)$, where $\mathbf{H} \in \mathcal{H} \subset \mathbb{R}^P$ is a collective set of filter parameters in all the manifold convolutional filters.

# 4. Generalization analysis of GNNs based on manifolds

We consider a manifold $\mathcal{M}$ as defined in Section 3.2, with a weighted Laplace operator $\mathcal{L}_\rho$ as defined in equation 3. Since functions $f \in L^2(\mathcal{M})$ characterize information over manifold $\mathcal{M}$, we restrict our analysis to a finite-dimensional subset of $L^2(\mathcal{M})$ up to some eigenvalue of $\mathcal{L}_\rho$, defined as a bandlimited signal.

**Definition 1.** *A manifold signal $f \in L^2(\mathcal{M})$ is bandlimited if there exists some $\lambda > 0$ such that for all eigenpairs $\{\lambda_i, \phi_i\}_{i=1}^\infty$ of the weighted Laplacian $\mathcal{L}_\rho$ when $\lambda_i > \lambda$, we have $\langle f, \phi_i \rangle_{\mathcal{M}} = 0$.*

Suppose we are given a set of $N$ i.i.d. randomly sampled points $X_N = \{x_i\}_{i=1}^N$ over $\mathcal{M}$, with $x_i \in \mathcal{M}$ sampled according to measure $\mu$. We construct a graph $\mathbf{G}(\mathcal{V}, \mathcal{E}, \mathcal{W})$ on these $N$ sampled points $X_N$, where each point $x_i$ is a vertex of graph $\mathbf{G}$, i.e. $\mathcal{V} = X_N$. Each pair of vertices $(x_i, x_j)$ is connected with an edge while the weight attached to the edge $\mathcal{W}(x_i, x_j)$ is determined by a kernel function $K_\epsilon$. The kernel function is decided by the Euclidean distance $\|x_i - x_j\|$ between these two points. The graph Laplacian denoted as $\mathbf{L}_N$ can be calculated based on the weight function (Merris, 1995). The constructed graph Laplacian with an appropriate kernel function has been proved to approximate the Laplace operator $\mathcal{L}_\rho$ of $\mathcal{M}$ (Calder & Trillos, 2022; Belkin & Niyogi, 2008; Dunson et al., 2021). We present the following two definitions of $K_\epsilon$.

**Definition 2** (Gaussian kernel based graph (Belkin & Niyogi, 2008)). *The graph $\mathbf{G}(X_N, \mathcal{E}, \mathcal{W})$ can be constructed in $(x_i, x_j) \in \mathcal{E}$, as a dense graph degree when the kernel function is defined as*

$$\mathcal{W}(x_i, x_j) = K_{\epsilon,1} \left( \frac{\|x_i - x_j\|^2}{\epsilon} \right) \qquad (7)$$

$$= \frac{1}{N} \frac{1}{\epsilon^{d/2+1}(4\pi)^{d/2}} e^{-\frac{\|x_i - x_j\|^2}{4\epsilon}}. \qquad (8)$$

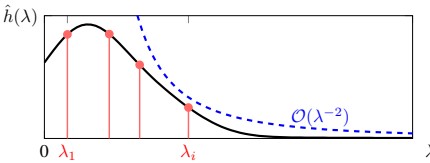

Figure 2: Frequency response illustration

The weight function of a Gaussian kernel based graph is defined on unbounded support (i.e. $[0, \infty)$), which connects $x_i$ and $x_j$ regardless of the distance between them. This results in a dense graph with $N^2$ edges. In particular, this Gaussian kernel based graph has been widely used to define the weight value function due to the good approximation properties of the corresponding graph Laplacians to the manifold Laplace operator (Dunson et al., 2021; Belkin & Niyogi, 2008; Xie et al., 2013).

**Definition 3** ($\epsilon$-graph (Calder & Trillos, 2022)). *The graph $\mathbf{G}(X_N, \mathcal{E}, \mathcal{W})$ can be constructed as an $\epsilon$-graph with the kernel function defined as*

$$\mathcal{W}(x_i, x_j) = K_{\epsilon, 2}\left(\frac{\|x_i - x_j\|^2}{\epsilon}\right) \tag{9}$$

$$= \frac{1}{N}\frac{d+2}{\epsilon^{d/2+1}\alpha_d}\mathbb{1}_{[0,1]}\left(\frac{\|x_i - x_j\|^2}{\epsilon}\right), \tag{10}$$

*with $(x_i, x_j) \in \mathcal{E}$, where $\alpha_d$ is the volume of a unit ball of dimension $d$ and $\mathbb{1}$ is the characteristic function.*

The weight function of an $\epsilon$-graph is defined on a bounded support, i.e., only nodes that are within a certain distance of one another can be connected by an edge. It has also been shown to provide a good approximation of the manifold Laplace operator (Calder & Trillos, 2022).

### 4.1. Manifold label prediction via node label prediction

Suppose we have an input manifold signal $f \in L^2(\mathcal{M})$ and a label (i.e. target) manifold signal $g \in L^2(\mathcal{M})$ over $\mathcal{M}$. With an MNN $\mathbf{\Phi}(\mathbf{H}, \mathcal{L}_\rho, \cdot)$, we predict the target value $g(x)$ based on input $f(x)$ at each point $x \in \mathcal{M}$. By sampling $N$ points $X_N$ over this manifold, we can approximate this problem in a discrete graph domain. Consider a graph $\mathbf{G}(X_N, \mathcal{E}, \mathcal{W})$ constructed with $X_N$ as either a Gaussian kernel based graph (Definition 2) or an $\epsilon$-graph (Definition 3) equipped with the graph Laplacian $\mathbf{L}_N$. Suppose we are given graph signal $\{\mathbf{x}, \mathbf{y}\}$ sampled from $\{f, g\}$ to train a GNN $\mathbf{\Phi_G}(\mathbf{H}, \mathbf{L}_N, \cdot)$, explicitly written as

$$[\mathbf{x}]_i = f(x_i), \qquad [\mathbf{y}]_i = g(x_i) \quad \text{for all } x_i \in X_N. \tag{11}$$

We assume that the filters in MNN $\mathbf{\Phi}(\mathbf{H}, \mathcal{L}_\rho, \cdot)$ and GNN $\mathbf{\Phi_G}(\mathbf{H}, \mathbf{L}_N, \cdot)$ satisfy a continuity assumption as follows, which is illustrated in Figure 2.

**AS 1.** *The frequency response function of the filter satisfies*

$$\left|\hat{h}(\lambda)\right| = \mathcal{O}\left(\lambda^{-d}\right), \quad \left|\hat{h}'(\lambda)\right| \leq C_L \lambda^{-d-1}, \quad \lambda \in (0, \infty), \tag{12}$$

*with $C_L$ a* spectral continuity constant *that regularizes the smoothness of the filter function.*

To introduce the first of our two main results, we require introducing two assumptions.

**AS 2.** *(Normalized Lipschitz nonlinearity) The nonlinearity $\sigma$ is normalized Lipschitz continuous, i.e., $|\sigma(a) - \sigma(b)| \leq |a - b|$, with $\sigma(0) = 0$.*

**AS 3.** *(Normalized Lipschitz loss function) The loss function $\ell$ is normalized Lipschitz continuous, i.e., $|\ell(y_i, y) - \ell(y_j, y)| \leq |y_i - y_j|$, with $\ell(y, y) = 0$.*

Assumption 2 is satisfied by most activations used in practice such as ReLU, modulus and sigmoid.

The generalization gap is evaluated between the *empirical risk* over the discrete graph model and the *statistical risk* over manifold model, with the manifold model viewed as a statistical model since the expectation of the sampled point is with respect to the measure $\mu$ over the manifold. The empirical risk over the sampled graph that we trained to minimize is therefore defined as

$$R_{\mathbf{G}}(\mathbf{H}) = \frac{1}{N}\sum_{i=1}^{N}\ell\left([\mathbf{\Phi_G}(\mathbf{H}, \mathbf{L}_N, \mathbf{x})]_i, [\mathbf{y}]_i\right). \tag{13}$$

The statistical risk over the manifold is defined as

$$R_{\mathcal{M}}(\mathbf{H}) = \int_{\mathcal{M}}\ell\left(\mathbf{\Phi}(\mathbf{H}, \mathcal{L}_\rho, f)(x), g(x)\right)\mathrm{d}\mu(x). \tag{14}$$

The generalization gap is defined to be

$$GA = \sup_{\mathbf{H} \in \mathcal{H}}|R_{\mathcal{M}}(\mathbf{H}) - R_{\mathbf{G}}(\mathbf{H})|. \tag{15}$$

**Theorem 1.** *Suppose the GNN and MNN with filters satisfying Assumption 1 have $L$ layers with $F$ features in each layer and the input signal is bandlimited (Definition 1). Under Assumptions 2 and 3 it holds in probability at least $1 - \delta$ that*

$$GA \leq F^L C_3 \left(\frac{\log N}{N}\right)^{\frac{1}{d}} \tag{16}$$

$$+ LF^{L-1}\left((C_1 C_L + C_2)\sqrt{\frac{\epsilon}{N}} + \frac{\pi^2\sqrt{\log(1/\delta)}}{6N}\right),$$

*when $d \geq 3$. If $d = 2$, the first term would be $F^L C_3 \frac{(\log N)^{3/4}}{N^{1/2}}$, with $C_1$, $C_2$, and $C_3$ depending on the geometry of $\mathcal{M}$, $C_L$ is the spectral continuity constant in Assumption 1.*

1. *When the graph is constructed with a Gaussian kernel equation 7, then $\epsilon \sim \left( \frac{\log(C/\delta)}{N} \right)^{\frac{2}{d+4}}$.*

2. *When the graph is constructed as an $\epsilon$-graph as equation 9, then $\epsilon \sim \left( \frac{\log(CN/\delta)}{N} \right)^{\frac{2}{d+4}}$.*

*Proof.* See Appendix D for proof and the definitions of $C_1$, $C_2$ and $C_3$. □

Theorem 1 shows that the generalization gap decreases approximately linearly with the number of nodes $N$ in the logarithmic scale, that is, $\log(GA) = \tilde{\mathcal{O}}(-\log N)$ with $\tilde{\mathcal{O}}$ as the $\mathcal{O}$ notation that ignores logarithmic orders, and that it also increases with the dimension of the underlying manifold $d$. Another observation is that the generalization gap scales with the size of the GNN architecture. Most importantly, we note the bound increases linearly with the spectral continuity constant $C_L$ (Assumption 1) – a smaller $C_L$ leads to a smaller generalization gap bound, and thus a better generalization capability. While a smaller $C_L$ leads to a smoother GNN, it discriminates fewer spectral components and, therefore, possesses worse discriminability. Consequently, we may observe a larger training loss with these smooth filters, as filters with worse discriminability encompass a smaller hypothesis function class and deteriorate the GNNs' approximation to the target functions during training. Since the testing loss can be upper bounded by the sum of training loss and the bound of generalization gap, on a smoother GNN (a smaller $C_L$), the performance on the training data will be closer to the performance on unseen testing data. Therefore, having a GNN with a smaller spectral continuity constant $C_L$ can guarantee more generalizable performance over unseen data from the same manifold. This also indicates that similar testing performance can be achieved by either a GNN with smaller training loss and worse generalization or a GNN with larger training loss and better generalization. In all, this indicates that there exists an optimal point to take the best advantage of the trade-off between a smaller generalization gap and better discriminability, resulting in a smaller testing loss decided by the spectral continuity constant of the GNN.

### 4.2. Manifold classification via graph classification

Suppose we have a set of manifolds $\{\mathcal{M}_k\}_{k=1}^K$, each of which is $d_k$-dimensional, smooth, compact, differentiable and embedded in $\mathbb{R}^M$ with measure $\mu_k$. Each manifold $\mathcal{M}_k$ equipped with a weighted Laplace operator $\mathcal{L}_{\rho_k,k}$ is labeled with $y_k \in \mathbb{R}$. We assume to have access to $N_k$ randomly sampled points according to measure $\mu_k$ over each manifold $\mathcal{M}_k$ and construct $K$ graphs $\{\mathbf{G}_k\}_{k=1}^K$ with graph Laplacians $\mathbf{L}_{N_k,k}$. The GNN $\boldsymbol{\Phi}_{\mathbf{G}_\cdot}(\mathbf{H}, \mathbf{L}_{N_\cdot,\cdot}, \mathbf{x}_\cdot)$ is trained on this set of graphs with $\mathbf{x}_k$ as the input graph signal sampled from the manifold signal $f_k \in L^2(\mathcal{M}_k)$ and

$y_k \in \mathbb{R}$ as the scalar target label. The final output of the GNN is set to be the average of the output signal values on each node while the output of MNN $\boldsymbol{\Phi}(\mathbf{H}, \mathcal{L}_{\rho,\cdot,\cdot}, f_\cdot)$ is the statistical average value of the output signal over the manifold. A loss function $\ell$ evaluates the difference between the output of GNN and MNN with the target label. The empirical risk of the GNN is

$$R_{\mathbf{G}}(\mathbf{H}) = \sum_{k=1}^{K} \ell \left( \frac{1}{N_k} \sum_{i=1}^{N_k} [\boldsymbol{\Phi}(\mathbf{H}, \mathbf{L}_{N_k,k}, \mathbf{x}_k)]_i, y_k \right).$$
(17)

While the output of MNN is the average value over the manifold, the statistical risk is defined based on the loss evaluated between the MNN output and the label as

$$R_{\mathcal{M}}(\mathbf{H}) = \sum_{k=1}^{K} \ell \left( \int_{\mathcal{M}_k} \boldsymbol{\Phi}(\mathbf{H}, \mathcal{L}_{\rho_k,k}, f_k)(x) \mathrm{d}\mu_k(x), y_k \right).$$
(18)

The generalization gap is therefore

$$GA = \sup_{\mathbf{H} \in \mathcal{H}} |R_{\mathcal{M}}(\mathbf{H}) - R_{\mathbf{G}}(\mathbf{H})|.$$
(19)

**Theorem 2.** *Suppose the GNN and MNN with filters satisfying Assumption 1 have $L$ layers with $F$ features in each layer and the input signal is bandlimited (Definition 1). Under Assumptions 2 and 3 it holds in probability at least $1 - \delta$ that*

$$GA \leq LF^{L-1} \sum_{k=1}^{K} (C_1 C_L + C_2) \left( \sqrt{\frac{\epsilon_k}{N_k}} \right.$$
(20)
$$\left. + \frac{\pi^2 \sqrt{\log(1/\delta)}}{6N_k} \right) + F^L C_3 \sum_{k=1}^{K} \left( \frac{\log N_k}{N_k} \right)^{\frac{1}{d_k}},$$

*when $d \geq 3$. If $d = 2$, the last term would be $F^L C_3 \sum_{k=1}^{K} \frac{(\log N_k)^{3/4}}{N_k^{1/2}}$, with $C_1$, $C_2$, and $C_3$ depending on the geometry of $\mathcal{M}$, $C_L$ is the spectal continuity constant in Assumption 1.*

1. *When the graphs are constructed with a Gaussian kernel equation 7, then $\epsilon_k \sim \left( \frac{\log(C/\delta)}{N_k} \right)^{\frac{2}{d_k+4}}$.*

2. *When the graphs are constructed as an $\epsilon$-graphs as equation 9, then $\epsilon_k \sim \left( \frac{\log(CN_k/\delta)}{N_k} \right)^{\frac{2}{d_k+4}}$.*

*Proof.* See Appendix F for proof and the definitions of $C_1$, $C_2$ and $C_3$. □

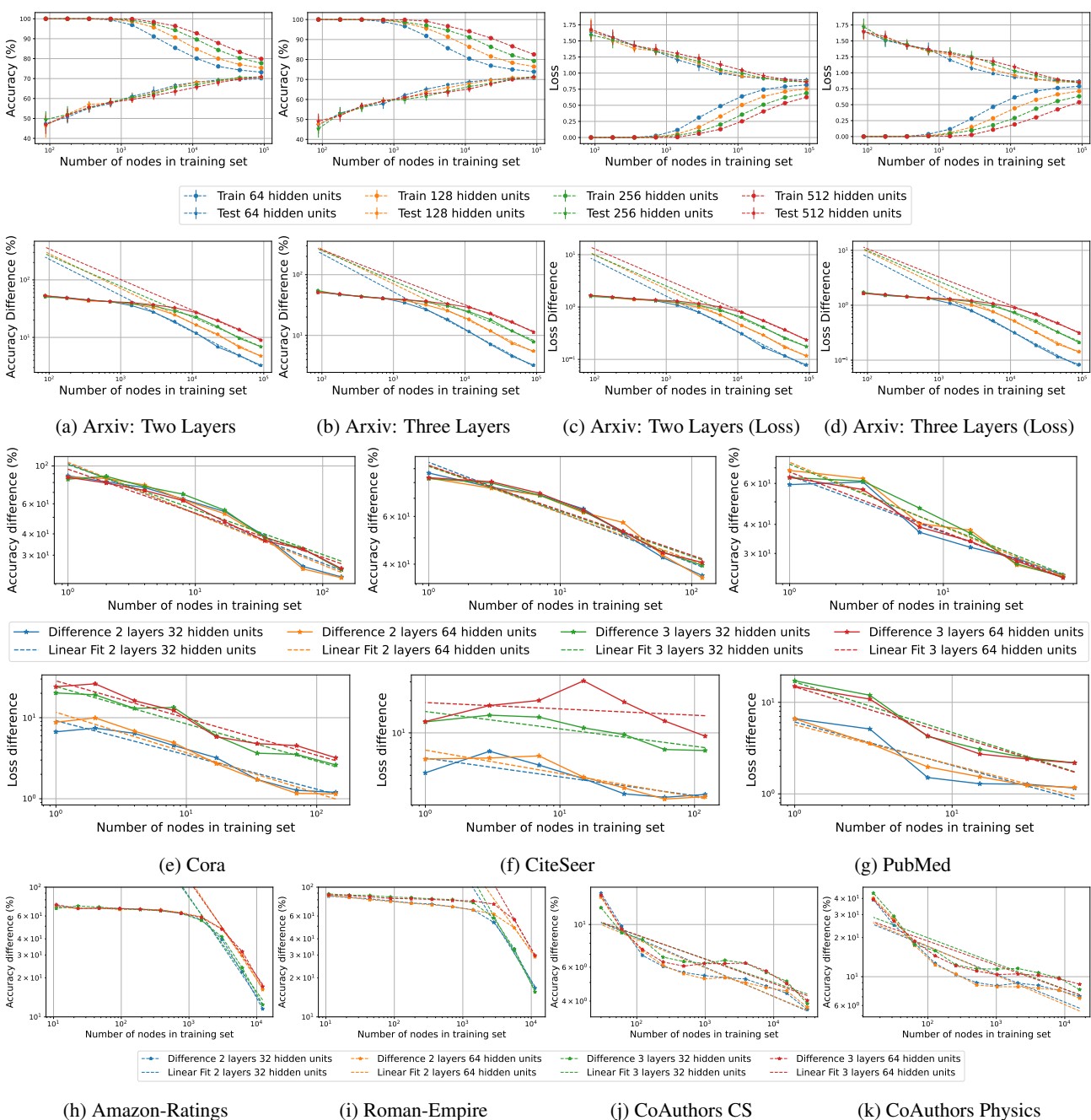

Figure 3: Merged visualization of all datasets: Arxiv (top row), Planetoid (middle row), and Heterophilic and CoAuthors datasets (bottom row). Each row provides accuracy and loss generalization gaps across different configurations and datasets.

Theorem 2 shows that a single graph sampled from the underlying manifold with large enough sampled points $N_k$ from each manifold $\mathcal{M}_k$ can provide an effective approximation to classify the manifold itself. The generalization gap also attests that the trained GNN can generalize to classify other unseen graphs sampled from the same manifold. Similar to the generalization result in node-level tasks, the generalization gap decreases with the number of points sampled over each manifold while increasing with the manifold dimension. A higher dimensional manifold, i.e. higher complexity, needs more samples to guarantee the generalization. The generalization gap also shows a trade-off between the generalization and discriminability as the bound increases linearly with the spectral continuity constant $C_L$. That is, to guarantee that a GNN for graph classification can generalize effectively, we must impose restrictions on the continuity of

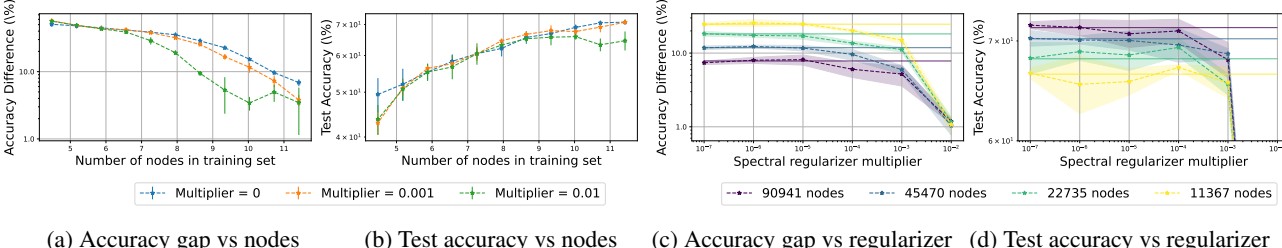

(a) Accuracy gap vs nodes     (b) Test accuracy vs nodes     (c) Accuracy gap vs regularizer   (d) Test accuracy vs regularizer

Figure 4: Spectral continuity constant effect on generalization gap and test accuracy.

its filter functions, which in turn limits the filters' ability to discriminate between different graph features.

We note that our assumption of a constant number of features can be generalized to include a different number of features in each layer for both node and graph classification.

## 5. Experiments

In this section, we empirically study the generalization gap in 8 real-world datasets. The task is to predict the label of a node given a set of features. The datasets vary in the number of nodes from $169,343$ to $3,327$, and in the number of edges from $1,166,243$ to $9,104$. The feature dimension also varies from $8,415$ to $300$ features, and the number of classes of the node label from $40$ to $3$. In all cases, we vary the number of nodes in the training set by partitioning it in $\{1, 2, 4, 8, 16, 32, 64, 32, 64, 128, 256, 512, 1024\}$ partitions when possible. For both the training and testing sets, we computed the loss in *cross-entropy loss*, and the accuracy in percentage ($\%$). Our main goal is to show that the rate presented in Theorem 1 holds in practice. In Figure 3, we plot the generalization gap of the accuracy in the logarithmic scale for a two-layered GNN (Figure 3a), and for a three-layered GNN (Figure 3b). On the upper side, we can see that the generalization bound decreases with the number of nodes and that outside of the strictly overfitting regime (when the training loss is below $95\%$), the generalization gap shows a linear decay, as depicted in the dashed line. The same behavior can be seen in Figures 3c, and 3d which correspond to the loss for 2 and 3 layered GNNs. As predicted by our theory, the generalization gap increases with the number of features and layers in the GNN. The behavior of the training and testing accuracy as a function of the number of nodes is intuitive. For the training loss, when the number of nodes in the training set is small, the GNN can overfit the training data. As the number of features increases, the GNN's capacity to overfit also increases. In Figures 3e to 3k, we present the accuracy generalization gaps for 2 and 3 layers with $32$ and $64$ features. In the overfitting regime, the rate of our generalization bound seems to hold – decreases linearly with the number of nodes in the logarithmic scale. In the non-overfitting regime, our rate holds for the points

whose training accuracy is below $95\%$. Also, we validate that the bound increases both with the number of features and the number of layers.

To measure the impact of the spectral continuity constant $C_L$, we add a regularizer to the cross-entropy loss (see Appendix J.3). We vary the value of the regularizer, noting that a larger regularizer translates into a smaller $C_L$ and therefore a smoother function. In Figures 4a and 4c we see the empirical manifestation of the bound that we showed (cf. Theorem 1) – a GNN with a smaller $C_L$ (a larger regularizer) will attain a smaller generalization gap. We can see that a larger regularizer (smaller continuity constant $C_L$, green line, regularizer $0.01$) attains a smaller generalization gap, and as the regularization decreases ($C_L$ increases), the generalization gap increases. The effect of having smaller spectral continuity constants $C_L$ is the lack of discriminability of the GNN. As can be seen in Figures 4b and 4d, the test error decreases when the multiplier is too large ($C_L$ too small). Therefore, a spectral regularize not too large can be shown to guarantee good test accuracy, but if the regularizer is too large, the test accuracy will be hurt by the lack of discriminability of the GNN as shown in Figure 4d. In all, we verify the fact that a GNN with a smoother spectral response will have a smaller generalization gap as shown in Theorem 1.

## 6. Conclusion

We study the statistical generalization of GNNs from a manifold perspective. We consider graphs sampled from manifolds and prove that GNNs could effectively generalize to unseen data from the manifolds when the number of sampled points is large enough and the filter functions are continuous in the spectral domain. We verify our theoretical results on both synthetic and real-world datasets. The impact of this paper is to show a better understanding of GNN generalization capabilities from a spectral perspective relying on a continuous model. Our work also motivates the practical design of large-scale GNNs. Specifically, in order to achieve a better generalization, it is essential to restrict the discriminability of GNNs by putting assumptions on the spectral continuity of the filter functions in the GNNs.

## Impact Statement

In this work, we explore the statistical generalization of GNNs from a manifold perspective by considering graphs sampled from manifolds. The impact of our work relies on showing that GNNs can effectively generalize to unseen data from the manifolds when the number of sampled points is large enough and the filter functions are continuous in the spectral domain. Our work also motivates the practical design of large-scale GNNs given that training on larger graphs attains a smaller generalization gap. Lastly, we observe that other than training on larger graphs, it is essential to restrict the discriminability of GNNs by putting assumptions on the spectral continuity of the filter functions in the GNNs.

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

# Contents

## A. Induced manifold signals

The graph signal attached to this constructed graph $\mathbf{G}$ can be seen as the discretization of the continuous function over the manifold. Suppose $f \in L^2(\mathcal{M})$, the graph signal $\mathbf{x}_N$ is composed of discrete data values of the function $f$ evaluated at $X_N$, i.e. $[\mathbf{x}_N]_i = f(x_i)$ for $i = 1, 2 \cdots, N$. With a sampling operator $\mathbf{P}_N : L^2(\mathcal{M}) \to L^2(X_N)$, the discretization can be written as

$$\mathbf{x}_N = \mathbf{P}_N f. \tag{21}$$

Let $\mu_N$ be the empirical measure of the random sample as

$$\mu_N = \frac{1}{N} \sum_{i=1}^{N} \delta_{x_i}. \tag{22}$$

Let $\{V_i\}_{i=1}^{N}$ be the decomposition (García Trillos et al., 2020) of $\mathcal{M}$ with respect to $X_N$ with $V_i \subset B_r(x_i)$, where $B_r(x_i)$ denotes the closed metric ball of radius $r$ centered at $x_i \in \mathcal{M}$ with respect to the Euclidean distance in the Euclidean ambient space. The decomposition can be achieved by the optimal transportation map $T : \mathcal{M} \to X_N$, which is defined by the $\infty$-Optimal Transport distance between $\mu$ and $\mu_N$.

$$d_\infty(\mu, \mu_N) := \min_{T: T_\# \mu = \mu_N} \mathrm{esssup}_{x \in \mathcal{M}} d(x, T(x)), \tag{23}$$

where $T_\# \mu = \mu_N$ indicates that $\mu(T^{-1}(V)) = \mu_N(V)$ for every $V_i$ of $\mathcal{M}$. This transportation map $T$ induces the partition $V_1, V_2, \cdots V_N$ of $\mathcal{M}$, where $V_i := T^{-1}(\{x_i\})$ with $\mu(V_i) = \frac{1}{N}$ for all $i = 1, \cdots N$. The radius of $V_i$ can be bounded as $r \leq A(\log N/N)^{1/d}$ when the manifold dimension $d \geq 3$ and $r \leq A(\log N)^{3/4}/N^{1/2}$ when $d = 2$ with $A$ related to the geometry of $\mathcal{M}$ (García Trillos et al., 2020)[Theorem 2].

The manifold function induced by the graph signal $\mathbf{x}_N$ over the sampled graph $\mathbf{G}$ is defined by

$$(\mathbf{I}_N \mathbf{x}_N)(x) = \sum_{i=1}^{N} [\mathbf{x}]_i \mathbb{1}_{x \in V_i}, \text{for all } x \in \mathcal{M} \tag{24}$$

where we denote $\mathbf{I}_N : L^2(X_N) \to L^2(\mathcal{M})$ as the inducing operator.

## B. Convergence of GNN to MNN

The convergence of GNN on sampled graphs to MNN provides the support for the generalization analysis. We first introduce the inner product over the manifold. The inner product of signals $f, g \in L^2(\mathcal{M})$ is defined as

$$\langle f, g \rangle_\mathcal{M} = \int_\mathcal{M} f(x)g(x)\mathrm{d}\mu(x), \tag{25}$$

where $\mathrm{d}\mu(x)$ is the volume element with respect to the measure $\mu$ over $\mathcal{M}$. Similarly, the norm of the manifold signal $f$ is

$$\|f\|_\mathcal{M}^2 = \langle f, f \rangle_\mathcal{M}. \tag{26}$$

**Proposition 1.** *Let $\mathcal{M} \subset \mathbb{R}^\mathsf{M}$ be an embedded manifold with weighted Laplace operator $\mathcal{L}_\rho$ and a bandlimited manifold signal $f$. Graph $\mathbf{G}_N$ is constructed based on a set of $N$ i.i.d. randomly sampled points $X_N = \{x_1, x_2, \cdots, x_N\}$ according to measure $\mu$ over $\mathcal{M}$. A graph signal $\mathbf{x}$ is the sampled manifold function values at $X_N$. The graph Laplacian $\mathbf{L}_N$ is calculated based on equation 7 or equation 9 with $\epsilon$ as the graph parameter. Let $\Phi(\mathbf{H}, \mathcal{L}_\rho, \cdot)$ be a MNN on $\mathcal{M}$ equation 6 with $L$ layers and $F$ features in each layer. Let $\Phi_\mathbf{G}(\mathbf{H}, \mathbf{L}_N, \cdot)$ be the GNN with the same architecture applied to the graph $\mathbf{G}_N$. Then, with the filters satisfy Assumption 1 and nonlinearities as normalized Lipschitz continuous, it holds in probability at least $1 - \delta$ that*

$$\frac{1}{N} \sum_{i=1}^{N} \|\Phi_\mathbf{G}(\mathbf{H}, \mathbf{L}_N, \mathbf{x}) - \mathbf{P}_N \Phi(\mathbf{H}, \mathcal{L}_\rho, \mathbf{I}_N \mathbf{x})\|_2 \leq LF^{L-1} \left( C_1 \sqrt{\epsilon} + C_2 \sqrt{\frac{\log(1/\delta)}{N}} \right) \tag{27}$$

*where $C_1, C_2$ are constants defined in the following proof.*

**Proposition 2.** *(Wang et al., 2024a)[Proposition 2, Proposition 4] Let $\mathcal{M} \subset \mathbb{R}^M$ be equipped with Laplace operator $\mathcal{L}_\rho$, whose eigendecomposition is given by $\{\lambda_i, \phi_i\}_{i=1}^\infty$. Let $\mathbf{L}_N$ be the discrete graph Laplacian of graph weights defined as equation 7 (or equation 9), with spectrum $\{\lambda_{i,N}, \phi_{i,N}\}_{i=1}^N$. Fix $K \in \mathbb{N}^+$ and assume that $\epsilon = \epsilon(N) \geq (\log(C/\delta)/N)^{2/(d+4)}$ (or $\epsilon = \epsilon(N) \geq (\log(CN/\delta)/N)^{2/(d+4)}$). Then, with probability at least $1 - \delta$, we have*

$$|\lambda_i - \lambda_{i,N}| \leq C_{\mathcal{M},1}\lambda_i\sqrt{\epsilon}, \quad \|a_i\phi_{i,N} - \phi_i\| \leq C_{\mathcal{M},2}\frac{\lambda_i}{\theta_i}\sqrt{\epsilon}, \tag{28}$$

*with $a_i \in \{-1, 1\}$ for all $i < K$ and $\theta$ the eigengap of $\mathcal{L}$, i.e., $\theta_i = \min\{\lambda_i - \lambda_{i-1}, \lambda_{i+1} - \lambda_i\}$. The constants $C_{\mathcal{M},1}$, $C_{\mathcal{M},2}$ depend on $d$ and the volume, the injectivity radius and sectional curvature of $\mathcal{M}$.*

*Proof.* Because $\{x_1, x_2, \cdots, x_N\}$ is a set of randomly sampled points from $\mathcal{M}$, based on Theorem 19 in (Von Luxburg et al., 2008) we can claim that

$$|\langle \mathbf{P}_N f, \mathbf{P}_N \phi_i\rangle - \langle f, \phi_i\rangle_{\mathcal{M}}| = O\left(\sqrt{\frac{\log(1/\delta)}{N}}\right). \tag{29}$$

This also indicates that

$$\left|\|\mathbf{P}_N f\|^2 - \|f\|_{\mathcal{M}}^2\right| = O\left(\sqrt{\frac{\log(1/\delta)}{N}}\right), \tag{30}$$

which indicates $\|\mathbf{P}_N f\| = \|f\|_{\mathcal{M}} + O((\log(1/\delta)/N)^{1/4})$. We suppose that the input manifold signal is $\lambda_M$-bandlimited with $M$ spectral components. We first write out the filter representation as

$$\|\mathbf{h}(\mathbf{L}_N)\mathbf{P}_N f - \mathbf{P}_N \mathbf{h}(\mathcal{L}_\rho)f\| = \left\|\sum_{i=1}^N \hat{h}(\lambda_{i,N})\langle \mathbf{P}_N f, \phi_{i,N}\rangle\phi_{i,N} - \sum_{i=1}^M \hat{h}(\lambda_i)\langle f, \phi_i\rangle_{\mathcal{M}}\mathbf{P}_N\phi_i\right\| \tag{31}$$

$$\leq \left\|\sum_{i=1}^M \hat{h}(\lambda_{i,N})\langle \mathbf{P}_N f, \phi_{i,N}\rangle\phi_{i,N} - \sum_{i=1}^M \hat{h}(\lambda_i)\langle f, \phi_i\rangle_{\mathcal{M}}\mathbf{P}_N\phi_i + \sum_{i=M+1}^N \hat{h}(\lambda_{i,N})\langle \mathbf{P}_N f, \phi_{i,N}\rangle\phi_{i,N}\right\| \tag{32}$$

$$\leq \left\|\sum_{i=1}^M \hat{h}(\lambda_{i,N})\langle \mathbf{P}_N f, \phi_{i,N}\rangle\phi_{i,N} - \sum_{i=1}^M \hat{h}(\lambda_i)\langle f, \phi_i\rangle_{\mathcal{M}}\mathbf{P}_N\phi_i\right\| + \left\|\sum_{i=M+1}^N \hat{h}(\lambda_{i,N})\langle \mathbf{P}_N f, \phi_{i,N}\rangle\phi_{i,N}\right\| \tag{33}$$

The first part of equation 33 can be decomposed with the triangle inequality as

$$\left\|\sum_{i=1}^M \hat{h}(\lambda_{i,N})\langle \mathbf{P}_N f, \phi_{i,N}\rangle\phi_{i,N} - \sum_{i=1}^M \hat{h}(\lambda_i)\langle f, \phi_i\rangle_{\mathcal{M}}\mathbf{P}_N\phi_i\right\|$$

$$\leq \left\|\sum_{i=1}^M \left(\hat{h}(\lambda_{i,N}) - \hat{h}(\lambda_i)\right)\langle \mathbf{P}_N f, \phi_{i,N}\rangle\phi_{i,N}\right\| + \left\|\sum_{i=1}^M \hat{h}(\lambda_i)\left(\langle \mathbf{P}_N f, \phi_{i,N}\rangle\phi_{i,N} - \langle f, \phi_i\rangle_{\mathcal{M}}\mathbf{P}_N\phi_i\right)\right\|. \tag{34}$$

In equation 34, the first part relies on the difference of eigenvalues and the second part depends on the eigenvector difference. The first term in equation 34 is bounded with Cauchy-Schwartz inequality as

$$\left\|\sum_{i=1}^M (\hat{h}(\lambda_{i,n}) - \hat{h}(\lambda_i))\langle \mathbf{P}_N f, \phi_{i,N}\rangle\phi_{i,N}\right\| \leq \sum_{i=1}^M \left|\hat{h}(\lambda_{i,N}) - \hat{h}(\lambda_i)\right||\langle \mathbf{P}_N f, \phi_{i,N}\rangle| \tag{35}$$

$$\leq \|\mathbf{P}_N f\|\sum_{i=1}^M |\hat{h}'(\lambda_i)||\lambda_{i,N} - \lambda_i| \tag{36}$$

$$\leq \|\mathbf{P}_N f\|\sum_{i=1}^M C_{\mathcal{M},1}C_L\sqrt{\epsilon}\lambda_i^{-d} \tag{37}$$

$$\leq \|\mathbf{P}_N f\|C_L C_{\mathcal{M},1}\sqrt{\epsilon}\sum_{i=1}^M i^{-2} \tag{38}$$

$$\leq \left(\|f\|_{\mathcal{M}} + \left(\frac{\log(1/\delta)}{N}\right)^{\frac{1}{4}}\right)C_{\mathcal{M},1}\sqrt{\epsilon}\frac{\pi^2}{6} := A_1(N) \tag{39}$$

In equation 37, it depends on the filter assumption in Assumption 1. In equation 38, we implement Weyl's law (Arendt et al., 2009) which indicates that eigenvalues of Laplace operator scales with the order $\lambda_i \sim i^{2/d}$. The last inequality comes from the fact that $\sum_{i=1}^{\infty} i^{-2} = \frac{\pi^2}{6}$. The second term in equation 34 can be bounded with the triangle inequality as

$$
\left\| \sum_{i=1}^{M} \hat{h}(\lambda_i) \left( \langle \mathbf{P}_N f, \phi_{i,N} \rangle \phi_{i,N} - \langle f, \phi_i \rangle_{\mathcal{M}} \mathbf{P}_N \phi_i \right) \right\|
$$
$$
\leq \left\| \sum_{i=1}^{M} \hat{h}(\lambda_i) \left( \langle \mathbf{P}_N f, \phi_{i,N} \rangle \phi_{i,N} - \langle \mathbf{P}_N f, \phi_{i,N} \rangle \mathbf{P}_N \phi_i \right) \right\|
$$
$$
+ \left\| \sum_{i=1}^{M} \hat{h}(\lambda_i) \left( \langle \mathbf{P}_N f, \phi_{i,N} \rangle \mathbf{P}_N \phi_i - \langle f, \phi_i \rangle_{\mathcal{M}} \mathbf{P}_N \phi_i \right) \right\| \tag{40}
$$

The first term in equation 40 can be bounded with inserting the eigenfunction convergence result in Proposition 2 as

$$
\left\| \sum_{i=1}^{M} \hat{h}(\lambda_i) \left( \langle \mathbf{P}_N f, \phi_{i,N} \rangle \phi_{i,N} - \langle \mathbf{P}_N f, \phi_{i,N} \rangle_{\mathcal{M}} \mathbf{P}_N \phi_i \right) \right\|
$$
$$
\leq \sum_{i=1}^{M} \left| \hat{h}(\lambda_i) \right| \|\mathbf{P}_N f\| \|\phi_{i,N} - \mathbf{P}_N \phi_i\| \tag{41}
$$
$$
\leq \sum_{i=1}^{M} (\lambda_i^{-d+1}) \frac{C_{\mathcal{M},2}\sqrt{\epsilon}}{\theta_i} \left( \|f\|_{\mathcal{M}} + \left( \frac{\log(1/\delta)}{N} \right)^{\frac{1}{4}} \right) \tag{42}
$$
$$
\leq \sum_{i=1}^{M} (\lambda_i^{-d+1}) \max_{i=1,\cdots,M} \theta_i^{-1} C_{\mathcal{M},2}\sqrt{\epsilon} \left( \|f\|_{\mathcal{M}} + \left( \frac{\log(1/\delta)}{N} \right)^{\frac{1}{4}} \right) \tag{43}
$$
$$
:= A_2(M, N). \tag{44}
$$

Considering the filter assumption in Assumption 1, the second term in equation 40 can be written as

$$
\left\| \sum_{i=1}^{M} \hat{h}(\lambda_{i,N}) (\langle \mathbf{P}_N f, \phi_{i,N} \rangle \mathbf{P}_N \phi_i - \langle f, \phi_i \rangle_{\mathcal{M}} \mathbf{P}_N \phi_i) \right\|
$$
$$
\leq \sum_{i=1}^{M} \left| \hat{h}(\lambda_{i,N}) \right| |\langle \mathbf{P}_N f, \phi_{i,N} \rangle - \langle f, \phi_i \rangle_{\mathcal{M}}| \|\mathbf{P}_N \phi_i\| \tag{45}
$$
$$
\leq \sum_{i=1}^{M} (\lambda_{i,N}^{-d}) |\langle \mathbf{P}_N f, \phi_{i,N} \rangle - \langle f, \phi_i \rangle_{\mathcal{M}}| \left( 1 + \left( \frac{\log(1/\delta)}{N} \right)^{\frac{1}{4}} \right) \tag{46}
$$
$$
\leq \sum_{i=1}^{M} (1 + C_{\mathcal{M},1}\sqrt{\epsilon})^{-d} (\lambda_i^{-d}) |\langle \mathbf{P}_N f, \phi_{i,N} \rangle - \langle f, \phi_i \rangle_{\mathcal{M}}| \left( 1 + \left( \frac{\log(1/\delta)}{N} \right)^{\frac{1}{4}} \right) \tag{47}
$$
$$
\leq \frac{\pi^2}{6} |\langle \mathbf{P}_N f, \phi_{i,N} \rangle - \langle f, \phi_i \rangle_{\mathcal{M}}| \left( 1 + \left( \frac{\log(1/\delta)}{N} \right)^{\frac{1}{4}} \right) := A_3(N) \tag{48}
$$

The term $|\langle \mathbf{P}_N f, \phi_{i,N} \rangle - \langle f, \phi_i \rangle_{\mathcal{M}}|$ can be decomposed by inserting a term $\langle \mathbf{P}_N f, \mathbf{P}_N \phi_i \rangle$ as

$$
|\langle \mathbf{P}_N f, \phi_{i,N} \rangle - \langle f, \phi_i \rangle_{\mathcal{M}}| \leq |\langle \mathbf{P}_N f, \phi_{i,N} \rangle - \langle \mathbf{P}_N f, \mathbf{P}_N \phi_i \rangle + \langle \mathbf{P}_N f, \mathbf{P}_N \phi_i \rangle - \langle f, \phi_i \rangle_{\mathcal{M}}| \tag{49}
$$
$$
\leq |\langle \mathbf{P}_N f, \phi_{i,N} \rangle - \langle \mathbf{P}_N f, \mathbf{P}_N \phi_i \rangle| + |\langle \mathbf{P}_N f, \mathbf{P}_N \phi_i \rangle - \langle f, \phi_i \rangle_{\mathcal{M}}| \tag{50}
$$
$$
\leq \|\mathbf{P}_N f\| \|\phi_{i,N} - \mathbf{P}_N \phi_i\| + |\langle \mathbf{P}_N f, \mathbf{P}_N \phi_i \rangle - \langle f, \phi_i \rangle_{\mathcal{M}}| \tag{51}
$$
$$
\leq \left( \|f\|_{\mathcal{M}} + \left( \frac{\log(1/\delta)}{N} \right)^{\frac{1}{4}} \right) \frac{C_{\mathcal{M},2}\lambda_i\sqrt{\epsilon}}{\theta_i} + \sqrt{\frac{\log(1/\delta)}{N}} \tag{52}
$$

Then equation equation 47 can be bounded as

$$\left\| \sum_{i=1}^{M} \hat{h}(\lambda_{i,N})(\langle \mathbf{P}_N f, \phi_{i,N} \rangle \mathbf{P}_N \phi_i - \langle f, \phi_i \rangle_{\mathcal{M}} \mathbf{P}_N \phi_i) \right\|$$

$$\leq \sum_{i=1}^{M} (1 + C_{\mathcal{M},1}\sqrt{\epsilon})^{-d}(\lambda_i^{-d}) \left( \left( \|f\|_{\mathcal{M}} + \left(\frac{\log(1/\delta)}{N}\right)^{\frac{1}{4}} \right) \frac{C_{\mathcal{M},2}\lambda_i\sqrt{\epsilon}}{\theta_i} + \sqrt{\frac{\log(1/\delta)}{N}} \right) \left( 1 + \left(\frac{\log(1/\delta)}{N}\right)^{\frac{1}{4}} \right) \quad (53)$$

$$\leq \frac{\pi^2}{6} \max_{i=1,\cdots,M} \frac{C_{\mathcal{M},2}\sqrt{\epsilon}}{\theta_i} \left( \|f\|_{\mathcal{M}} + \left(\frac{\log(1/\delta)}{N}\right)^{\frac{1}{4}} \right) + \frac{\pi^2}{6} \sqrt{\frac{\log(1/\delta)}{N}} \quad (54)$$

The second term in equation 33 can be bounded with the eigenvalue difference bound in Proposition 2 as

$$\left\| \sum_{i=M+1}^{N} \hat{h}(\lambda_{i,N})\langle \mathbf{P}_N f, \phi_{i,N} \rangle \phi_{i,N} \right\| \leq \sum_{i=M+1}^{N} (\lambda_{i,N}^{-d}) \left( \|f\|_{\mathcal{M}} + \left(\frac{\log(1/\delta)}{N}\right)^{\frac{1}{4}} \right) \quad (55)$$

$$\leq \sum_{i=M+1}^{\infty} (\lambda_{i,N}^{-d})\|f\|_{\mathcal{M}} \quad (56)$$

$$\leq (1 + C_{\mathcal{M},1}\sqrt{\epsilon})^{-d} \sum_{i=M+1}^{\infty} (\lambda_i^{-d})\|f\|_{\mathcal{M}} \quad (57)$$

$$\leq M^{-1}\|f\|_{\mathcal{M}} := A_4(M). \quad (58)$$

We note that the bound is made up by terms $A_1(N) + A_2(M,N) + A_3(N) + A_4(M)$, related to the bandwidth of manifold signal $M$ and the number of sampled points $N$. This makes the bound scale with the order

$$\|\mathbf{h}(\mathbf{L}_N)\mathbf{P}_N f - \mathbf{P}_N \mathbf{h}(\mathcal{L}_\rho)f\| \leq C_1'\sqrt{\epsilon} + C_2'\sqrt{\epsilon}\theta_M^{-1} + C_3'\sqrt{\frac{\log(1/\delta)}{N}} + C_4'M^{-1}, \quad (59)$$

with $C_1' = C_L C_{\mathcal{M},1}\frac{\pi^2}{6}\|f\|_{\mathcal{M}}$, $C_2' = C_{\mathcal{M},2}\frac{\pi^2}{6}$, $C_3' = \frac{\pi^2}{6}$ and $C_4' = \|f\|_{\mathcal{M}}$. As $N$ goes to infinity, for every $\delta > 0$, there exists some $M_0$, such that for all $M > M_0$ it holds that $A_4(M) \leq \delta/2$. There also exists $n_0$, such that for all $N > n_0$, it holds that $A_1(N) + A_2(M_0,N) + A_3(N) \leq \delta/2$. We can conclude that the summations converge as $N$ goes to infinity. We see $M$ large enough to have $M^{-1} \leq \delta'$, which makes the eigengap $\theta_M$ also bounded by some constant. We combine the first two terms as

$$\|\mathbf{h}(\mathbf{L}_N)\mathbf{P}_N f - \mathbf{P}_N \mathbf{h}(\mathcal{L}_\rho)f\| \leq (C_1 C_L + C_2)\sqrt{\epsilon} + \frac{\pi^2}{6}\sqrt{\frac{\log(1/\delta)}{N}}, \quad (60)$$

with $C_1 = C_{\mathcal{M},1}\frac{\pi^2}{6}\|f\|_{\mathcal{M}}$ and $C_2 = C_{\mathcal{M},2}\frac{\pi^2}{6}\theta_{\delta'^{-1}}^{-1}$. To bound the output difference of MNNs, we need to write in the form of features of the final layer

$$\|\mathbf{\Phi_G}(\mathbf{H},\mathbf{L}_N,\mathbf{P}_N f) - \mathbf{P}_N\mathbf{\Phi}(\mathbf{H},\mathcal{L}_\rho,f))\| = \left\| \sum_{q=1}^{F}\mathbf{x}_{n,L}^q - \sum_{q=1}^{F}\mathbf{P}_N f_L^q \right\| \leq \sum_{q=1}^{F}\left\| \mathbf{x}_{n,L}^q - \mathbf{P}_N f_L^q \right\|. \quad (61)$$

By inserting the definitions, we have

$$\left\| \mathbf{x}_{n,l}^p - \mathbf{P}_N f_l^p \right\| = \left\| \sigma\left(\sum_{q=1}^{F}\mathbf{h}_l^{pq}(\mathbf{L}_N)\mathbf{x}_{n,l-1}^q\right) - \mathbf{P}_N\sigma\left(\sum_{q=1}^{F}\mathbf{h}_l^{pq}(\mathcal{L}_\rho)f_{l-1}^q\right) \right\| \quad (62)$$

with $\mathbf{x}_{n,0} = \mathbf{P}_N f$ as the input of the first layer. With a normalized point-wise Lipschitz nonlinearity, we have

$$\|\mathbf{x}_{n,l}^p - \mathbf{P}_N f_l^p\| \leq \left\| \sum_{q=1}^{F}\mathbf{h}_l^{pq}(\mathbf{L}_N)\mathbf{x}_{n,l-1}^q - \mathbf{P}_N\sum_{q=1}^{F}\mathbf{h}_l^{pq}(\mathcal{L}_\rho)f_{l-1}^q \right\| \quad (63)$$

$$\leq \sum_{q=1}^{F}\left\| \mathbf{h}_l^{pq}(\mathbf{L}_N)\mathbf{x}_{n,l-1}^q - \mathbf{P}_N\mathbf{h}_l^{pq}(\mathcal{L}_\rho)f_{l-1}^q \right\| \quad (64)$$

The difference can be further decomposed as

$$\|\mathbf{h}_l^{pq}(\mathbf{L}_N)\mathbf{x}_{n,l-1}^q - \mathbf{P}_N\mathbf{h}_l^{pq}(\mathcal{L}_\rho)f_{l-1}^q\|$$

$$\leq \|\mathbf{h}_l^{pq}(\mathbf{L}_N)\mathbf{x}_{n,l-1}^q - \mathbf{h}_l^{pq}(\mathbf{L}_N)\mathbf{P}_N f_{l-1}^q + \mathbf{h}_l^{pq}(\mathbf{L}_N)\mathbf{P}_N f_{l-1}^q - \mathbf{P}_N\mathbf{h}_l^{pq}(\mathcal{L}_\rho)f_{l-1}^q\| \tag{65}$$

$$\leq \left\|\mathbf{h}_l^{pq}(\mathbf{L}_N)\mathbf{x}_{n,l-1}^q - \mathbf{h}_l^{pq}(\mathbf{L}_N)\mathbf{P}_N f_{l-1}^q\right\| + \left\|\mathbf{h}_l^{pq}(\mathbf{L}_N)\mathbf{P}_N f_{l-1}^q - \mathbf{P}_N\mathbf{h}_l^{pq}(\mathcal{L}_\rho)f_{l-1}^q\right\| \tag{66}$$

The second term can be bounded with equation 59 and we denote the bound as $\Delta_N$ for simplicity. The first term can be decomposed by Cauchy-Schwartz inequality and non-amplifying of the filter functions as

$$\left\|\mathbf{x}_{n,l}^p - \mathbf{P}_N f_l^p\right\| \leq \sum_{q=1}^{F}\Delta_N\|\mathbf{x}_{n,l-1}^q\| + \sum_{q=1}^{F}\|\mathbf{x}_{l-1}^q - \mathbf{P}_N f_{l-1}^q\|. \tag{67}$$

To solve this recursion, we need to compute the bound for $\|\mathbf{x}_l^p\|$. By normalized Lipschitz continuity of $\sigma$ and the fact that $\sigma(0) = 0$, we can get

$$\|\mathbf{x}_l^p\| \leq \left\|\sum_{q=1}^{F}\mathbf{h}_l^{pq}(\mathbf{L}_N)\mathbf{x}_{l-1}^q\right\| \leq \sum_{q=1}^{F}\|\mathbf{h}_l^{pq}(\mathbf{L}_N)\|\,\|\mathbf{x}_{l-1}^q\| \leq \sum_{q=1}^{F}\|\mathbf{x}_{l-1}^q\| \leq F^{l-1}\|\mathbf{x}\|. \tag{68}$$

Insert this conclusion back to solve the recursion, we can get

$$\left\|\mathbf{x}_{n,l}^p - \mathbf{P}_N f_l^p\right\| \leq lF^{l-1}\Delta_N\|\mathbf{x}\|. \tag{69}$$

Replace $l$ with $L$ we can obtain

$$\|\boldsymbol{\Phi}_{\mathbf{G}}(\mathbf{H}, \mathbf{L}_N, \mathbf{P}_N f) - \mathbf{P}_N\boldsymbol{\Phi}(\mathbf{H}, \mathcal{L}_\rho, f))\| \leq LF^{L-1}\Delta_N, \tag{70}$$

when the input graph signal is normalized. By replacing $f = \mathbf{I}_N\mathbf{x}$, we can conclude the proof. $\qquad\square$

## C. Local Lipschitz continuity of MNNs

We propose that the outputs of MNN defined in equation 6 are locally Lipschitz continuous within a certain area, which is stated explicitly as follows.

**Proposition 3.** *(Local Lipschitz continuity of MNNs) Assume that the assumptions in Theorem 1 hold. Let MNN be $L$ layers with $F$ features in each layer, suppose the manifold filters are nonamplifying with $|\hat{h}(\lambda)| \leq 1$ and the nonlinearities normalized Lipschitz continuous, then there exists a constant $C'$ such that*

$$|\boldsymbol{\Phi}(\mathbf{H}, \mathcal{L}_\rho, f)(x) - \boldsymbol{\Phi}(\mathbf{H}, \mathcal{L}_\rho, f)(y)| \leq F^L C' dist(x - y), \quad \text{for all } x, y \in B_r(\mathcal{M}), \tag{71}$$

*where $B_r(\mathcal{M})$ is a ball with radius $r$ over $\mathcal{M}$ with respect to the geodesic distance.*

*Proof.* The output of MNN can be written explicitly as

$$|\boldsymbol{\Phi}(\mathbf{H}, \mathcal{L}_\rho, f)(x) - \boldsymbol{\Phi}(\mathbf{H}, \mathcal{L}_\rho, f)(y)| = \left|\sigma\left(\sum_{q=1}^{F}\mathbf{h}_L^q(\mathcal{L}_\rho)f_{L-1}^q(x)\right) - \sigma\left(\sum_{q=1}^{F}\mathbf{h}_L^q(\mathcal{L}_\rho)f_{L-1}^q(y)\right)\right| \tag{72}$$

$$\leq \left|\sum_{q=1}^{F}\mathbf{h}_L^q(\mathcal{L}_\rho)f_{L-1}^q(x) - \sum_{q=1}^{F}\mathbf{h}_L^q(\mathcal{L}_\rho)f_{L-1}^q(y)\right| \leq F\max_{q=1,\cdots,F}\left|\mathbf{h}_L^q(\mathcal{L}_\rho)f_{L-1}^q(x) - \mathbf{h}_L^q(\mathcal{L}_\rho)f_{L-1}^q(y)\right|. \tag{73}$$

We have $f_{L-1}^q(x) = \sigma\left(\sum_{p=1}^{F}\mathbf{h}_{L-1}^p f_{L-2}^p(x)\right)$. The process can be repeated recursively by expanding $f_{L-1}^q(x)$ and $f_{L-1}^q(y)$, and finally, we can have

$$|\boldsymbol{\Phi}(\mathbf{H}, \mathcal{L}_\rho, f)(x) - \boldsymbol{\Phi}(\mathbf{H}, \mathcal{L}_\rho, f)(y)| \leq F^L|\mathbf{h}_L(\mathcal{L}_\rho)\cdots\mathbf{h}_1(\mathcal{L}_\rho)f(x) - \mathbf{h}_L(\mathcal{L}_\rho)\cdots\mathbf{h}_1(\mathcal{L}_\rho)f(y)|. \tag{74}$$

With $f$ as a $\lambda$-bandlimited manifold signal, we suppose $g = \mathbf{h}_L(\mathcal{L}_\rho) \cdots \mathbf{h}_1(\mathcal{L}_\rho)f$. As $\langle f, \phi_i \rangle = 0$ for all $i > M$, $g$ is also bandlimited and possesses $M$ spectral components. The gradient can be bounded according to (Shi & Xu, 2010) combined with the non-amplifying property of the filter function as

$$\|\nabla g\|_\infty \leq C \sum_{\lambda_i \leq \lambda} \left| \hat{h}(\lambda_i) \right|^L \lambda_i^{\frac{d+1}{2}} \|f\|_{\mathcal{M}} \leq C \sum_{\lambda_i \leq \lambda} \lambda_i^{\frac{d+1}{2}} \|f\|_{\mathcal{M}} \tag{75}$$

From Theorem 4.5 in (Evans, 2018), $g$ is locally Lipschitz continuous as

$$|g(x) - g(y)| \leq C' \mathrm{dist}(x - y), \quad \text{with } x, y \in B_r(\mathcal{M}), \tag{76}$$

where $B_r(\mathcal{M})$ is a closed ball with radius $r$ with $C'$ depending on the geometry of $\mathcal{M}$.

Combining the above, we have the continuity of the output of MNN as

$$|\mathbf{\Phi}(\mathbf{H}, \mathcal{L}_\rho, f)(x) - \mathbf{\Phi}(\mathbf{H}, \mathcal{L}_\rho, f)(y)| \leq F^L C' \mathrm{dist}(x - y), \quad \text{with } x, y \in B_r(\mathcal{M}), \tag{77}$$

which concludes the proof. $\qquad\square$

## D. Proof of Theorem 1

*Proof.* To analyze the difference between the empirical risk and statistical risk, we introduce an intermediate term which is the induced version of the sampled MNN output. We define $\mathbf{I}_N$ as the inducing operator based on the decomposition $\{V_i\}_{i=1}^N$ defined in Section A. This intermediate term is written explicitly as

$$\overline{\boldsymbol{\Phi}}(\mathbf{H}, \mathcal{L}_\rho, f)(x) = \mathbf{I}_N \mathbf{P}_N \boldsymbol{\Phi}(\mathbf{H}, \mathcal{L}_\rho, f)(x) = \sum_{i=1}^N \boldsymbol{\Phi}(\mathbf{H}, \mathcal{L}_\rho, f)(x_i) \mathbb{1}_{x \in V_i}, \text{ for all } x \in \mathcal{M}, \tag{78}$$

where $x_i \in X_N$ are sampled points from the manifold.

Suppose $\mathbf{H} \in \arg\min_{\mathbf{H} \in \mathcal{H}} R_{\mathcal{M}}(\mathbf{H})$, we have

$$GA = \sup_{\mathbf{H} \in \mathcal{H}} |R_{\mathbf{G}}(\mathbf{H}) - R_{\mathcal{M}}(\mathbf{H})| \tag{79}$$

The difference between $R_{\mathbf{G}}(\mathbf{H})$ and $R_{\mathcal{M}}(\mathbf{H})$ can be decomposed as

$$|R_{\mathbf{G}}(\mathbf{H}) - R_{\mathcal{M}}(\mathbf{H})|$$

$$= \left| \frac{1}{N} \sum_{i=1}^N \ell([\boldsymbol{\Phi}_{\mathbf{G}}(\mathbf{H}, \mathbf{L}_N, \mathbf{x})]_i, [\mathbf{y}]_i) - \int_{\mathcal{M}} \ell\left(\boldsymbol{\Phi}(\mathbf{H}, \mathcal{L}_\rho, f)(x), g(x)\right) d\mu(x) \right| \tag{80}$$

$$= \left| \frac{1}{N} \sum_{i=1}^N \ell([\boldsymbol{\Phi}_{\mathbf{G}}(\mathbf{H}, \mathbf{L}_N, \mathbf{x})]_i, [\mathbf{y}]_i) - \int_{\mathcal{M}} \ell\left(\overline{\boldsymbol{\Phi}}(\mathbf{H}, \mathcal{L}_\rho, f)(x), g(x)\right) d\mu(x) \right.$$

$$\left. + \int_{\mathcal{M}} \ell\left(\overline{\boldsymbol{\Phi}}(\mathbf{H}, \mathcal{L}_\rho, f)(x), g(x)\right) d\mu(x) - \int_{\mathcal{M}} \ell\left(\boldsymbol{\Phi}(\mathbf{H}, \mathcal{L}_\rho, f)(x), g(x)\right) d\mu(x) \right| \tag{81}$$

$$\leq \left| \frac{1}{N} \sum_{i=1}^N \ell([\boldsymbol{\Phi}_{\mathbf{G}}(\mathbf{H}, \mathbf{L}_N, \mathbf{x})]_i, [\mathbf{y}]_i) - \int_{\mathcal{M}} \ell\left(\overline{\boldsymbol{\Phi}}(\mathbf{H}, \mathcal{L}_\rho, f)(x), g(x)\right) d\mu(x) \right|$$

$$+ \left| \int_{\mathcal{M}} \ell\left(\overline{\boldsymbol{\Phi}}(\mathbf{H}, \mathcal{L}_\rho, f)(x), g(x)\right) d\mu(x) - \int_{\mathcal{M}} \ell\left(\boldsymbol{\Phi}(\mathbf{H}, \mathcal{L}_\rho, f)(x), g(x)\right) d\mu(x) \right| \tag{82}$$

We analyze the two terms in equation 82 separately, with the first term bounded based on the convergence of GNN to MNN and the second term bounded with the smoothness of manifold functions.

The first term in equation 82 can be written as

$$\left| \frac{1}{N} \sum_{i=1}^N \ell([\boldsymbol{\Phi}_{\mathbf{G}}(\mathbf{H}, \mathbf{L}_N, \mathbf{x})]_i, [\mathbf{y}]_i) - \int_{\mathcal{M}} \ell\left(\overline{\boldsymbol{\Phi}}(\mathbf{H}, \mathcal{L}_\rho, f)(x), g(x)\right) d\mu(x) \right| \tag{83}$$

$$= \frac{1}{N} \left| \sum_{i=1}^N \ell([\boldsymbol{\Phi}_{\mathbf{G}}(\mathbf{H}, \mathbf{L}_N, \mathbf{x})]_i, [\mathbf{y}]_i) - \sum_{i=1}^N \ell(\boldsymbol{\Phi}(\mathbf{H}, \mathcal{L}_\rho, f)(x_i), g(x_i)) \right| \tag{84}$$

$$\leq \frac{1}{N} \sum_{i=1}^N |\ell([\boldsymbol{\Phi}_{\mathbf{G}}(\mathbf{H}, \mathbf{L}_N, \mathbf{x})]_i, [\mathbf{y}]_i) - \ell(\boldsymbol{\Phi}(\mathbf{H}, \mathcal{L}_\rho, f)(x_i), g(x_i))| \tag{85}$$

$$\leq \frac{1}{N} \sum_{i=1}^N \left| [\boldsymbol{\Phi}_{\mathbf{G}}(\mathbf{H}, \mathbf{L}_N, \mathbf{x})]_i - \boldsymbol{\Phi}(\mathbf{H}, \mathcal{L}_\rho, f)(x_i) \right| \tag{86}$$

$$\leq \frac{1}{N} \|\boldsymbol{\Phi}_{\mathbf{G}}(\mathbf{H}, \mathbf{L}_N, \mathbf{x}) - \mathbf{P}_N \boldsymbol{\Phi}(\mathbf{H}, \mathcal{L}_\rho, \mathbf{I}_N \mathbf{x})\|_1 \tag{87}$$

$$\leq \frac{1}{\sqrt{N}} L F^{L-1} \left( (C_1 C_L + C_2)\sqrt{\epsilon} + \frac{\pi^2}{6} \sqrt{\frac{\log(1/\delta)}{N}} \right) \tag{88}$$

From equation 83 to equation 84, we use the definition of induced manifold signal defined in equation 78. We utilize the Lipschitz continuity assumption on loss function from equation 85 to equation 86. From equation 86 to equation 87, it

depends on the fact that $\mathbf{x}$ is a single-entry vector and that $[\mathbf{y}]_i$ is the value sampled from target manifold function $g$ evaluated on $x_i$. Finally the bound depends on the convergence of GNN on the sampled graph to the MNN as stated in Proposition 1.

The second term is decomposed as

$$\left| \int_{\mathcal{M}} \ell\left(\overline{\boldsymbol{\Phi}}(\mathbf{H}, \mathcal{L}_\rho, f)(x), g(x)\right) \mathrm{d}\mu(x) - \int_{\mathcal{M}} \ell\left(\boldsymbol{\Phi}(\mathbf{H}, \mathcal{L}_\rho, f)(x), g(x)\right) \mathrm{d}\mu(x) \right| \tag{89}$$

$$\leq \left| \sum_{i=1}^{N} \int_{V_i} \ell\left(\overline{\boldsymbol{\Phi}}(\mathbf{H}, \mathcal{L}_\rho, f)(x), g(x)\right) \mathrm{d}\mu(x) - \sum_{i=1}^{N} \int_{V_i} \ell\left(\boldsymbol{\Phi}(\mathbf{H}, \mathcal{L}_\rho, f)(x), g(x)\right) \mathrm{d}\mu(x) \right| \tag{90}$$

$$\leq \sum_{i=1}^{N} \int_{V_i} \left| \ell\left(\overline{\boldsymbol{\Phi}}(\mathbf{H}, \mathcal{L}_\rho, f)(x), g(x)\right) - \ell\left(\boldsymbol{\Phi}(\mathbf{H}, \mathcal{L}_\rho, f)(x), g(x)\right) \right| \mathrm{d}\mu(x) \tag{91}$$

$$\leq \sum_{i=1}^{N} \int_{V_i} \left| \overline{\boldsymbol{\Phi}}(\mathbf{H}, \mathcal{L}_\rho, f)(x) - \boldsymbol{\Phi}(\mathbf{H}, \mathcal{L}_\rho, f)(x) \right| \mathrm{d}\mu(x) \tag{92}$$

$$\leq \sum_{i=1}^{N} \int_{V_i} \left| \boldsymbol{\Phi}(\mathbf{H}, \mathcal{L}_\rho, f)(x_i) - \boldsymbol{\Phi}(\mathbf{H}, \mathcal{L}_\rho, f)(x) \right| \mathrm{d}\mu(x) \tag{93}$$

From equation 89 to equation 90, it relies on the decomposition of the MNN output over $\{V_i\}_{i=1}^N$. From equation 91 to equation 92, we use the Lipschitz continuity of loss function. From equation 92 to equation 93, we use the definition of $\overline{\boldsymbol{\Phi}}(\mathbf{H}, \mathcal{L}_\rho, f)$. Proposition 3 indicates that the MNN outputs are Lipschitz continuous within a certain range, which leads to

$$\sum_{i=1}^{N} \int_{V_i} \left| \boldsymbol{\Phi}(\mathbf{H}, \mathcal{L}_\rho, f)(x_i) - \boldsymbol{\Phi}(\mathbf{H}, \mathcal{L}_\rho, f)(x) \right| \mathrm{d}\mu(x)$$

$$\leq \sum_{i=1}^{N} \int_{V_i} F^L C_3 \left( \frac{\log N}{N} \right)^{\frac{1}{d}} \mathrm{d}\mu(x) \tag{94}$$

$$= F^L C_3 \left( \frac{\log N}{N} \right)^{\frac{1}{d}} \sum_{i=1}^{N} \int_{V_i} \mathrm{d}\mu(x) \tag{95}$$

$$\leq F^L C_3 \left( \frac{\log N}{N} \right)^{\frac{1}{d}}, \tag{96}$$

when $d \geq 3$. If $d = 2$, the bound would be

$$\sum_{i=1}^{N} \int_{V_i} \left| \boldsymbol{\Phi}(\mathbf{H}, \mathcal{L}_\rho, f)(x_i) - \boldsymbol{\Phi}(\mathbf{H}, \mathcal{L}_\rho, f)(x) \right| \mathrm{d}\mu(x) \leq F^L C_3 \frac{(\log N)^{\frac{3}{4}}}{N^{\frac{1}{2}}}. \tag{97}$$

Combining equation 88 and equation 96 (or equation 97), we can conclude the proof. □

# E. Corollary of Theorem 1

**Corollary 1.** *Suppose the GNN with filters satisfying Assumption 1 have L layers with F features in each layer and the input signal is bandlimited (Definition 1)). Suppose graphs $\mathbf{G}_1$ with $N_1$ nodes and $\mathbf{G}_2$ with $N_2$ nodes are sampled from the same underlying manifold $\mathcal{M}$. Under Assumptions 2 and 3 it holds in probability at least $1 - \delta$ that*

$$\sup_{\mathbf{H} \in \mathcal{H}} \left| \frac{1}{N_1} \sum_{i=1}^{N_1} \ell\left([\mathbf{\Phi}_{\mathbf{G}_1}(\mathbf{H}, \mathbf{L}_{N_1}, \mathbf{x}_1)]_i, [\mathbf{y}_1]_i\right) - \frac{1}{N_2} \sum_{i=1}^{N_2} \ell\left([\mathbf{\Phi}_{\mathbf{G}_2}(\mathbf{H}, \mathbf{L}_{N_2}, \mathbf{x}_2)]_i, [\mathbf{y}_2]_i\right) \right| \leq$$

$$LF^{L-1} \left( (C_1 C_L + C_2) \frac{\sqrt{\epsilon}(\sqrt{N_1} + \sqrt{N_2})}{\sqrt{N_1 N_2}} + \frac{\pi^2 (N_1 + N_2)\sqrt{\log(1/\delta)}}{6 N_1 N_2} \right) + F^L C_3 \left( \frac{\log N_1}{N_1} \right)^{\frac{1}{d}} + F^L C_3 \left( \frac{\log N_2}{N_2} \right)^{\frac{1}{d}},$$

$$(98)$$

*with $C_1$, $C_2$, and $C_3$ depending on the geometry of $\mathcal{M}$, $C_L$ is the spectral continuity constant in Assumption 1.*

*Proof.* By importing the statistical risk over the manifold $R_{\mathcal{M}}(\mathbf{H})$ in equation 14, the bound can be derived with a triangle inequality as

$$\sup_{\mathbf{H} \in \mathcal{H}} \left| \frac{1}{N_1} \sum_{i=1}^{N_1} \ell\left([\mathbf{\Phi}_{\mathbf{G}_1}(\mathbf{H}, \mathbf{L}_{N_1}, \mathbf{x}_1)]_i, [\mathbf{y}_1]_i\right) - \frac{1}{N_2} \sum_{i=1}^{N_2} \ell\left([\mathbf{\Phi}_{\mathbf{G}_2}(\mathbf{H}, \mathbf{L}_{N_2}, \mathbf{x}_2)]_i, [\mathbf{y}_2]_i\right) \right|$$

$$= \sup_{\mathbf{H} \in \mathcal{H}} \left| \frac{1}{N_1} \sum_{i=1}^{N_1} \ell\left([\mathbf{\Phi}_{\mathbf{G}_1}(\mathbf{H}, \mathbf{L}_{N_1}, \mathbf{x}_1)]_i, [\mathbf{y}_1]_i\right) - R_{\mathcal{M}}(\mathbf{H}) + R_{\mathcal{M}}(\mathbf{H}) - \frac{1}{N_2} \sum_{i=1}^{N_2} \ell\left([\mathbf{\Phi}_{\mathbf{G}_2}(\mathbf{H}, \mathbf{L}_{N_2}, \mathbf{x}_2)]_i, [\mathbf{y}_2]_i\right) \right| \quad (99)$$

$$\leq \sup_{\mathbf{H} \in \mathcal{H}} \left| \frac{1}{N_1} \sum_{i=1}^{N_1} \ell\left([\mathbf{\Phi}_{\mathbf{G}_1}(\mathbf{H}, \mathbf{L}_{N_1}, \mathbf{x}_1)]_i, [\mathbf{y}_1]_i\right) - R_{\mathcal{M}}(\mathbf{H}) \right| + \sup_{\mathbf{H} \in \mathcal{H}} \left| \frac{1}{N_2} \sum_{i=1}^{N_2} \ell\left([\mathbf{\Phi}_{\mathbf{G}_2}(\mathbf{H}, \mathbf{L}_{N_2}, \mathbf{x}_2)]_i, [\mathbf{y}_2]_i\right) - R_{\mathcal{M}}(\mathbf{H}) \right|.$$

$$(100)$$

Inserting the result in Theorem 1 concludes the proof. $\qquad\square$

## F. Proof of Theorem 2

*Proof.* We can write the difference as

$$|R_{\mathbf{G}}(\mathbf{H}) - R_{\mathcal{M}}(\mathbf{H})|$$

$$\leq \sum_{k=1}^{K} \left| \ell\left( \frac{1}{N} \sum_{i=1}^{N} [\mathbf{\Phi}_{\mathbf{G}}(\mathbf{H}, \mathbf{L}_{N,k}, \mathbf{x}_k)]_i, y_k \right) - \ell\left( \int_{\mathcal{M}_k} \mathbf{\Phi}(\mathbf{H}, \mathcal{L}_{\rho,k}, f_k) \mathrm{d}\mu_k(x), y_k \right) \right| \tag{101}$$

Based on the property of absolute value inequality and the Lipschitz continuity assumption of loss function (Assumption 3), we have

$$\left| \ell\left( \frac{1}{N} \sum_{i=1}^{N} [\mathbf{\Phi}_{\mathbf{G}}(\mathbf{H}, \mathbf{L}_{N,k}, \mathbf{x}_k)]_i, y_k \right) - \ell\left( \int_{\mathcal{M}_k} \mathbf{\Phi}(\mathbf{H}, \mathcal{L}_{\rho,k}, f_k) \mathrm{d}\mu_k(x), y_k \right) \right|$$

$$\leq \left| \frac{1}{N} \sum_{i=1}^{N} [\mathbf{\Phi}_{\mathbf{G}}(\mathbf{H}, \mathbf{L}_{N,k}, \mathbf{x}_k)]_i - \int_{\mathcal{M}_k} \mathbf{\Phi}(\mathbf{H}, \mathcal{L}_{\rho,k}, f_k) \mathrm{d}\mu_k(x) \right| \tag{102}$$

We insert an intermediate term $\mathbf{\Phi}(\mathbf{H}, \mathcal{L}_{\rho,k}, f_k)(x_i)$ as the value evaluated on the sampled point $x_i$, which leads to

$$\left| \frac{1}{N} \sum_{i=1}^{N} [\mathbf{\Phi}_{\mathbf{G}}(\mathbf{H}, \mathbf{L}_{N,k}, \mathbf{x}_k)]_i - \int_{\mathcal{M}_k} \mathbf{\Phi}(\mathbf{H}, \mathcal{L}_{\rho,k}, f_k) \mathrm{d}\mu_k(x) \right| \tag{103}$$

$$\leq \left| \frac{1}{N} \sum_{i=1}^{N} [\mathbf{\Phi}_{\mathbf{G}}(\mathbf{H}, \mathbf{L}_{N,k}, \mathbf{x}_k)]_i - \frac{1}{N} \sum_{i=1}^{N} \mathbf{\Phi}(\mathbf{H}, \mathcal{L}_{\rho,k}, f_k)(x_i) \right| +$$

$$\left| \frac{1}{N} \sum_{i=1}^{N} \mathbf{\Phi}(\mathbf{H}, \mathcal{L}_{\rho,k}, f_k)(x_i) - \int_{\mathcal{M}_k} \mathbf{\Phi}(\mathbf{H}, \mathcal{L}_{\rho,k}, f_k) \mathrm{d}\mu_k(x) \right| \tag{104}$$

The first term in equation 104 can be bounded similarly as equation 87, which is explicitly written as

$$\left| \frac{1}{N} \sum_{i=1}^{N} [\mathbf{\Phi}_{\mathbf{G}}(\mathbf{H}, \mathbf{L}_{N,k}, \mathbf{x}_k)]_i - \frac{1}{N} \sum_{i=1}^{N} \mathbf{\Phi}(\mathbf{H}, \mathcal{L}_{\rho,k}, f_k)(x_i) \right| \tag{105}$$

$$\leq \frac{1}{N} \|\mathbf{\Phi}_{\mathbf{G}}(\mathbf{H}, \mathbf{L}_N, \mathbf{x}_k) - \mathbf{P}_N \mathbf{\Phi}(\mathbf{H}, \mathcal{L}_{\rho}, f_k)\|_1 \tag{106}$$

$$\leq \frac{1}{\sqrt{N}} \|\mathbf{\Phi}_{\mathbf{G}}(\mathbf{H}, \mathbf{L}_N, \mathbf{x}_k) - \mathbf{P}_N \mathbf{\Phi}(\mathbf{H}, \mathcal{L}_{\rho}, f_k)\|_2 \tag{107}$$

$$\leq \frac{1}{\sqrt{N}} \left( (C_1 C_L + C_2)\sqrt{\epsilon} + \frac{\pi^2}{6} \sqrt{\frac{\log(1/\delta)}{N}} \right) \tag{108}$$

The second term is

$$\left| \frac{1}{N} \sum_{i=1}^{N} \mathbf{\Phi}(\mathbf{H}, \mathcal{L}_{\rho,k}, f_k)(x_i) - \int_{\mathcal{M}_k} \mathbf{\Phi}(\mathbf{H}, \mathcal{L}_{\rho,k}, f_k) \mathrm{d}\mu_k(x) \right| \tag{109}$$

$$= \left| \sum_{i=1}^{N} \int_{V_i} \mathbf{\Phi}(\mathbf{H}, \mathcal{L}_{\rho,k}, f_k)(x_i) \mathrm{d}\mu_k(x) - \sum_{i=1}^{N} \int_{V_i} \mathbf{\Phi}(\mathbf{H}, \mathcal{L}_{\rho,k}, f_k)(x) \mathrm{d}\mu_k(x) \right| \tag{110}$$

$$\leq \sum_{i=1}^{N} \int_{V_i} |\mathbf{\Phi}(\mathbf{H}, \mathcal{L}_{\rho,k}, f_k)(x_i) - \mathbf{\Phi}(\mathbf{H}, \mathcal{L}_{\rho,k}, f_k)(x)| \, \mathrm{d}\mu_k(x) \tag{111}$$

$$\leq F^L C_3 \left( \frac{\log N}{N} \right)^{\frac{1}{d}} \tag{112}$$

This depends on the Lipschitz continuity of the output manifold function in Proposition 3. □

## G. Further references

**Graphon theory**    Different from the manifold model we are using, some research constructs graphs derived from graphons, which can be viewed as a random limit graph model. This research has focused on their convergence, stability, as well as transferability (Ruiz et al., 2020; Maskey et al., 2023; Keriven et al., 2020). In (Parada-Mayorga et al., 2023), a graphon is used as a pooling tool in GNNs. Despite its utility, the graphon presents several limitations compared to the manifold model we use. Firstly, the graphon model assumes an infinite degree at every node (Lovász, 2012), which is not the case in the manifold model. Additionally, graphons offer limited insight into the underlying model; visualizing a graphon is challenging, except in the stochastic block model case. Manifolds, however, are more interpretable, especially when based on familiar shapes like spheres and 3D models (see Figures **??** and **??**). Finally, the manifold model supports a wider range of characterizable models, making it a more realistic choice.

**Transferability of GNNs**    The transferability of GNNs has been extensively studied by examining the differences in GNN outputs across graphs of varying sizes as they converge to a limit model. This analysis, however, often lacks statistical generalization. Several studies have explored GNN transferability with graphon models, proving bounds on the differences in GNN outputs (Ruiz et al., 2023; 2020; Maskey et al., 2023). Other research has demonstrated how increasing graph size during GNN training can improve generalization to large-scale graphs (Cervino et al., 2023). The transferability of GNNs has also been investigated in the context of graphs generated from general topological spaces (Levie et al., 2021) and manifolds (Wang et al., 2024a). Furthermore, a novel graphop operator has been proposed as a limit model for both dense and sparse graphs, with proven transferability results (Le & Jegelka, 2024). Further research has focused on transfer learning for GNNs by measuring distances between graphs without assuming a limit model (Lee et al., 2017; Zhu et al., 2021). Finally, a transferable graph transformer has been proposed and empirically validated (He et al., 2023).

## H. Filter Assumption

In the main results, we assume that the filters in GNN and MNN satisfy Assumption 1. This may lead to limited discriminability in high-frequency spectrum. While this is a reasonable assumption, high-frequency signals on graphs or manifolds can fluctuate significantly between adjacent entries, leading to instability and learning challenges. We expect a degree of local homogeneity, which translates to low-frequency signals. This assumption is supported by empirical evidence in various domains, including opinion dynamics, econometrics, and graph signal processing (Degroot, 1974; Billio et al., 2012; Ramakrishna et al., 2020). Moreover, several other effective learning techniques, such as Principal Component Analysis (PCA) and Isomap, implicitly employ low-pass filtering. Therefore, we believe that the filter assumption is not restrictive and is well-supported by both practical applications and theoretical considerations.

## I. Manifold Assumption

In this paper, we considered the case in which graphs are sampled from manifolds. This is an assumption that has been widely used in practice. From dynamical systems (Talmon et al., 2015) to images (Peyré, 2009; Osher et al., 2017), assuming an underlying low dimensional manifold is a common practice. Real-world graphs, like the ones considered in the node prediction experiments, can be assumed to be sampled from $d$-dimensional manifolds. To support this argument, in Figure 5, we plot the 100 largest eigenvalues of the Laplacian matrix associated with each graph. By doing this, we show a fast decay in the values of the eigenvalues progress. This decay shows that the information is mostly supported on a subset of the eigenvalues thus reinforcing the idea that it comes from a low dimensional manifold.

## J. Experiment details and further experiments

We consider the following datasets: *OGBN-Arxiv* (Wang et al., 2020; Mikolov et al., 2013), *Cora* (Yang et al., 2016b), *CiteSeer* (Yang et al., 2016b), *PubMed* (Yang et al., 2016b), *Coauthors CS* (Shchur et al., 2018), *Coauthors Physics* (Shchur et al., 2018), *Amazon-rating* (Platonov et al., 2023), and *Roman-Empire* (Platonov et al., 2023), details of the datasets can be found in Table 1.

All experiments were done using a *NVIDIA GeForce RTX 3090*, and each set of experiments took at most 10 hours to complete. In total, we run 10 datasets, which amounts for around 100 hours of GPU use. All datasets used in this paper are public, and free to use. They can be downloaded using the *pytorch* package (`https://pytorch-geometric.readthedocs.io/en/latest/modules/datasets.html`), the *ogb* package (`https://ogb.stanford.`

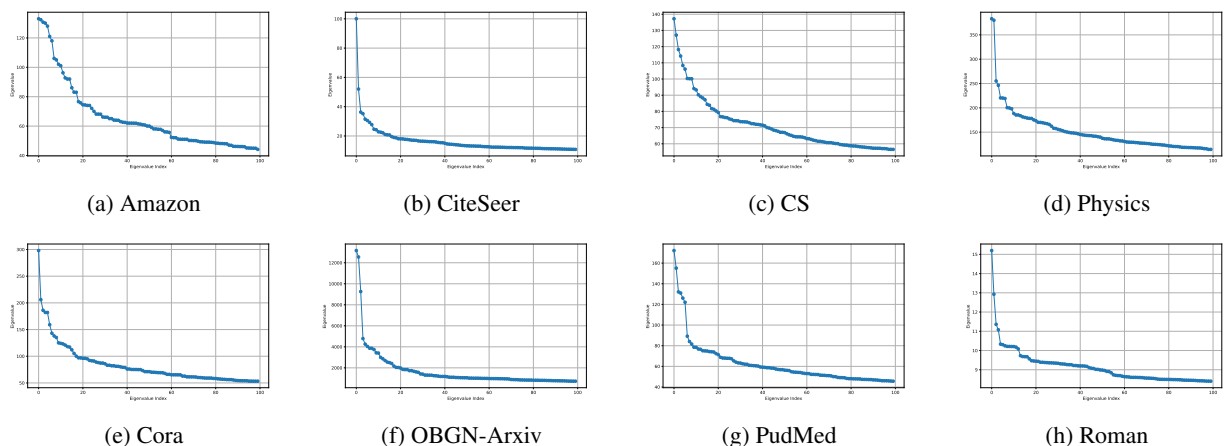

(a) Amazon      (b) CiteSeer      (c) CS      (d) Physics

(e) Cora      (f) OBGN-Arxiv      (g) PudMed      (h) Roman

Figure 5: Top 100 eigenvalues of the graph for each dataset considered in the node classification problem.

edu/docs/nodeprop/) and the Princeton ModelNet project (https://modelnet.cs.princeton.edu/). In total, the datasets occupy around 5 gb. However, they do not need to be all stored at the same time, as the experiments that we run can be done in series.

### J.1. ModelNet10 and ModelNet40 graph classification tasks

ModelNet10 dataset (Wu et al., 2015) includes 3,991 meshed CAD models from 10 categories for training and 908 models for testing as Figure 6 shows. ModelNet40 dataset includes 38,400 training and 9,600 testing models as Figure 7 shows. In each model, $N$ points are uniformly randomly selected to construct graphs to approximate the underlying model, such as chairs, tables.

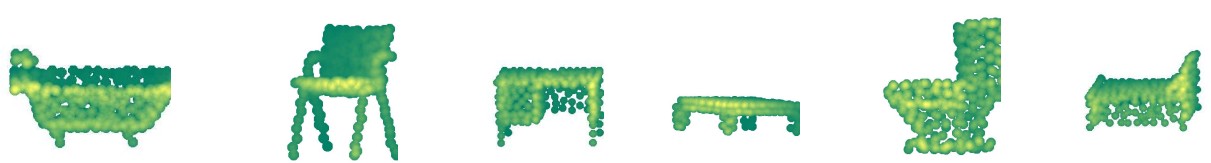

Figure 6: Point cloud models in ModelNet10 with $N = 300$ sampled points in each model, corresponding to bathtub, chair, desk, table, toiler, and bed.

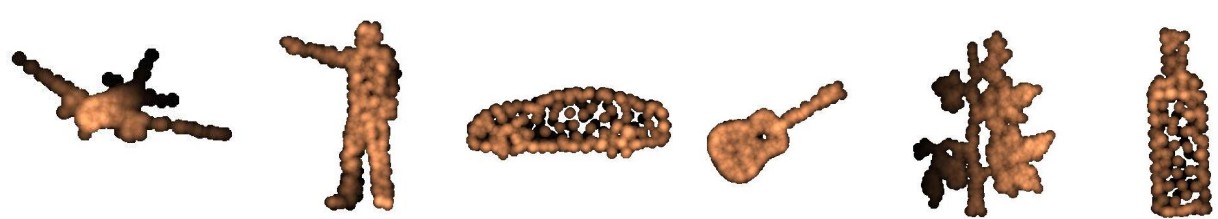

Figure 7: Point cloud models from ModelNet40 with $N = 300$ sampled points in each model, corresponding to airplane, person, car, guitar, plant, and bottle.

The weight function of the constructed graph is determined as equation 7 with $\epsilon = 0.1$. We calculate the Laplacian matrix for each graph as the input graph shift operator. In this experiment, we implement GNNs with different numbers of layers

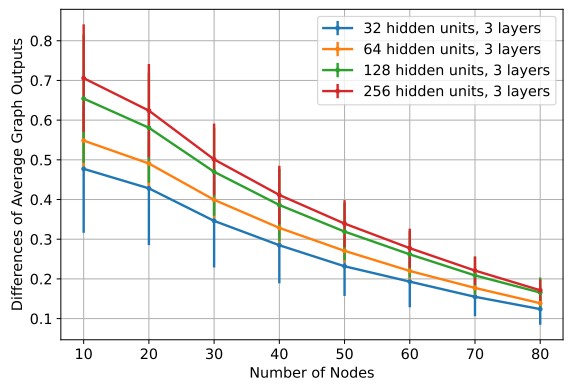
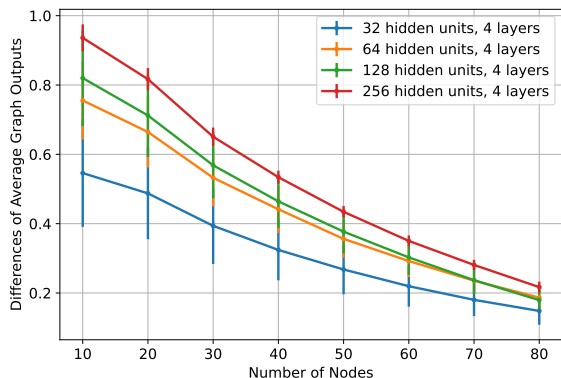

(a) Differences of the outputs of 3-layer GNNs.

(b) Differences of the outputs of 4-layer GNNs.

Figure 8: Graph outputs differences of GNNs with different architectures on ModelNet40 dataset.

and hidden units with $K = 5$ filters in each layer. All the GNN architectures are trained by minimizing the cross-entropy loss. We implement an ADAM optimizer with the learning rate set as 0.005 along with the forgetting factors 0.9 and 0.999. We carry out the training for 40 epochs with the size of batches set as 10. We run 5 random dataset partitions and show the average performances and the standard deviation across these partitions.

### J.2. Node classification training details and datasets

In this section, we present the results for node classification. In this paragraph we present the common details for all datasets, we will next delve into each specific detail inside the dataset subsection that follows.

| Name | Nodes | Edges | Features | Number of Classes | Reference |
|---|---|---|---|---|---|
| Arxiv | $169,343$ | $1,166,243$ | $128$ | $40$ | (Wang et al., 2020; Mikolov et al., 2013) |
| Cora | $2,708$ | $10,556$ | $1,433$ | $7$ | (Yang et al., 2016b) |
| CiteSeer | $3,327$ | $9,104$ | $3,703$ | $6$ | (Yang et al., 2016b) |
| PubMed | $19,717$ | $88,648$ | $500$ | $3$ | (Yang et al., 2016b) |
| Coauthor Physics | $18,333$ | $163,788$ | $6,805$ | $15$ | (Shchur et al., 2018) |
| Coauthor CS | $34,493$ | $495,924$ | $8,415$ | $5$ | (Shchur et al., 2018) |
| Amazon-ratings | $24,492$ | $93,050$ | $300$ | $5$ | (Platonov et al., 2023) |
| Roman-empire | $22,662$ | $32,927$ | $300$ | $18$ | (Platonov et al., 2023) |

Table 1: Details of the datasets considered in the experiments.

In all datasets, we used the graph convolutional layer `GCN`, and trained for 1000 epochs. For the optimizer, we used `AdamW`, with using a learning rate of $0.01$, and $0$ weight decay. We trained using the graph convolutional layer, with a varying number of layers and hidden units. For dropout, we used $0.5$. We trained using the cross-entropy loss. In all cases, we trained 2 and 3 layered GNNs.

To compute the linear approximation in the plots, we used the mean squared error estimator of the form

$$\mathbf{y} = s * \log(\mathbf{n}) + p. \tag{113}$$

Where $s$ is the slope, $p$ is the point, and $\mathbf{n}$ is the vector with the nodes in the training set for each experiment. Note that we repeated each experiment for 10 independent runs. In all experiments, we compute the value of $s$ and $p$ that minimize the mean square error over the mean of the experiment runs, and we compute the Pearson correlation index over those values.

Our experiment shows that our bound shows the same rate dependency as the experiments. That is to say, in the logarithmic scale, the generalization gap of GNNs is linear with respect to the logarithm of the number of nodes. In most cases, the

Pearson correlation index is above $0.9$ in absolute value, which indicates a strong linear relationship. We noticed that the linear relationship changes the slope in the overfitting regime, and in the non-overfitting regime. That is to say, when the GNN is overfitting the training set, the generalization gap decreases at a much slower rate than it does with the GNN does not have the capacity to do so. Therefore, in the case in which the GNN overfits the training set for all nodes when computed $s$ using all the samples in the experiment. On the other hand, when the number of nodes is large enough that the GNN cannot overfit the training set, then we computed the $s$ and $p$ with the nodes in the non overfitting regime.

### J.3. Spectral Continuity Constant Regularizer

We add a regularization term to the loss to better control the value of the spectral continuity constant (defined in Assumption 1) while training. To do so, given a convolutional filter $\mathbf{h} \in \mathbb{R}^K$, its associated spectral continuity constant is

$$R(\mathbf{h}) = \sum_{k=0}^{K-1} k|h_k|\lambda_{max}^{k-1}, \tag{114}$$

Where $\lambda_{max}$ is the largest eigenvalue of the graph $\mathbf{G}$.

#### J.3.1. ARXIV DATASET

For this datasets, we trained $2, 3, 4$ layered GNN. We also used a learning rate scheduler `ReduceLROnPlateau` with mode min, factor $0.5$, patience $100$ and a minimum learning rate of $0.001$.

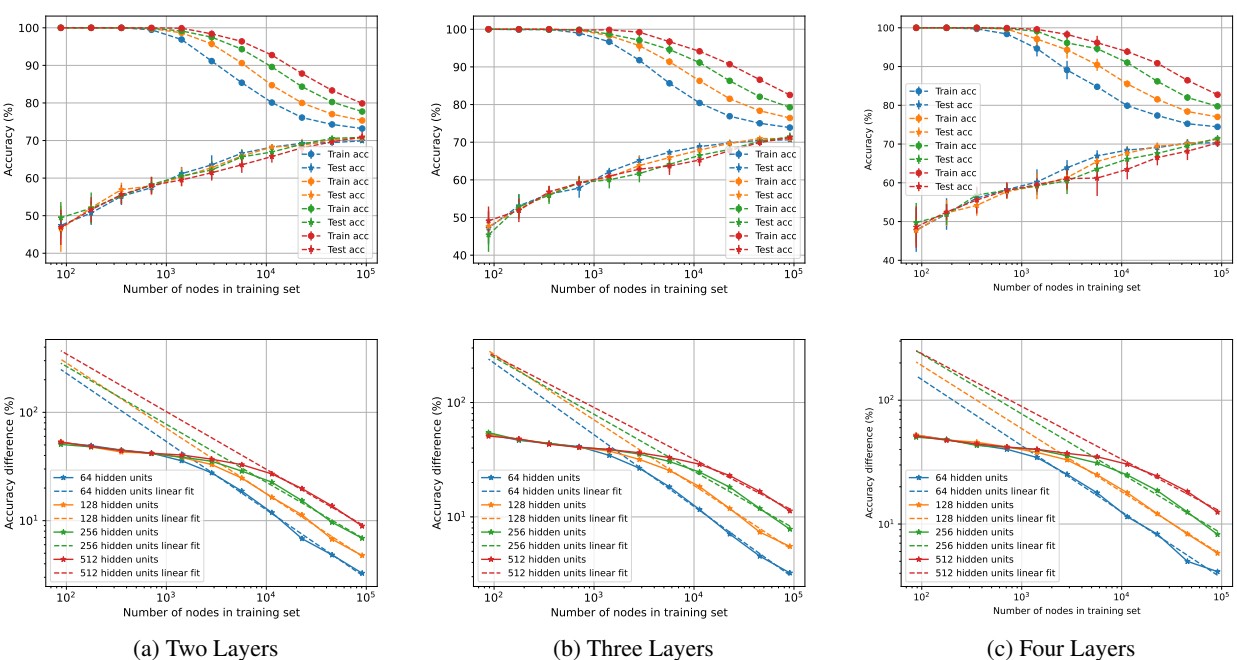

(a) Two Layers          (b) Three Layers          (c) Four Layers

Figure 9: Generalization gap for the OGBN-Arxiv dataset on the accuracy as a function of the number of nodes in the training set.

#### J.3.2. CORA DATASET

For the Cora dataset, we used the standard one, which can be obtained running `torch_geometric.datasets.Planetoid(root="./data",name='Cora')`.

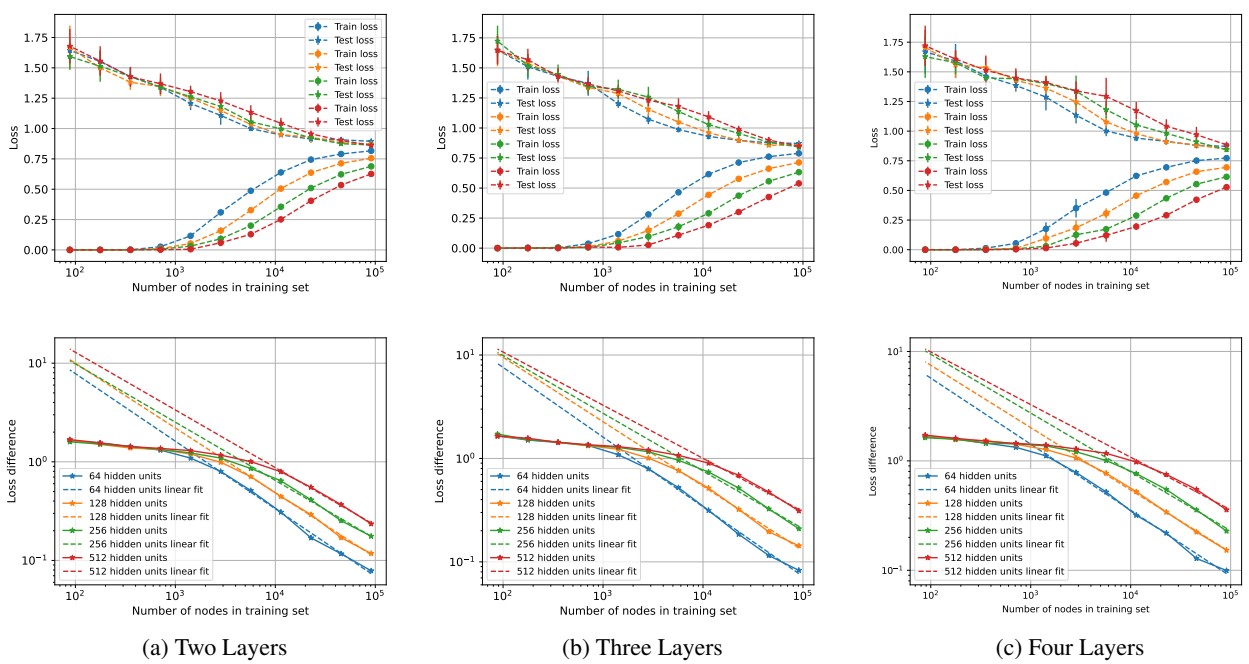

Figure 10: Generalization gap for the OGBN-arxiv dataset on the loss (cross-entropy) as a function of the number of nodes in the training set.

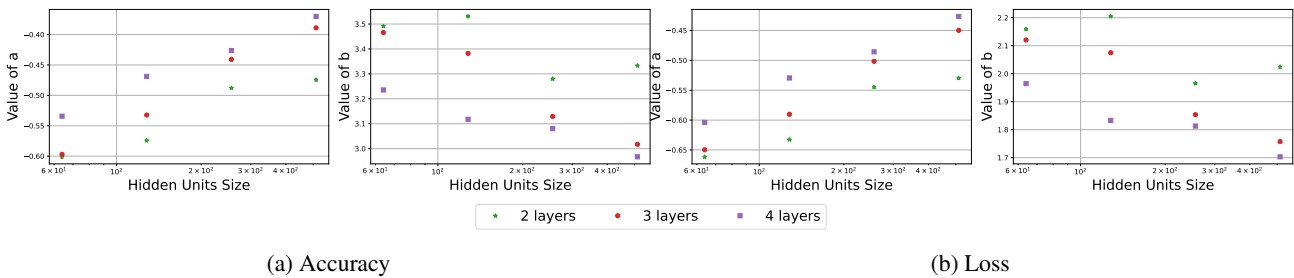

Figure 11: Values of slope (a) and point (b) corresponding to the linear fit ($a * \log(N) + b$) of Figures 10 and 9.

| Type | Lay. | Feat. | Slope | Point | Pearson Correlation Coefficient |
|---|---|---|---|---|---|
| Accuracy | 2 | 64 | $-6.301e-01$ | $3.621e+00$ | $-9.980e-01$ |
| Accuracy | 2 | 128 | $-6.034e-01$ | $3.663e+00$ | $-9.985e-01$ |
| Accuracy | 2 | 256 | $-5.347e-01$ | $3.493e+00$ | $-9.952e-01$ |
| Accuracy | 2 | 512 | $-5.328e-01$ | $3.605e+00$ | $-9.975e-01$ |
| Accuracy | 3 | 64 | $-6.271e-01$ | $3.600e+00$ | $-9.987e-01$ |
| Accuracy | 3 | 128 | $-5.730e-01$ | $3.567e+00$ | $-9.970e-01$ |
| Accuracy | 3 | 256 | $-4.986e-01$ | $3.393e+00$ | $-9.910e-01$ |
| Accuracy | 3 | 512 | $-4.529e-01$ | $3.315e+00$ | $-9.934e-01$ |
| Accuracy | 4 | 64 | $-5.343e-01$ | $3.236e+00$ | $-9.971e-01$ |
| Accuracy | 4 | 128 | $-5.096e-01$ | $3.299e+00$ | $-9.987e-01$ |
| Accuracy | 4 | 256 | $-4.827e-01$ | $3.337e+00$ | $-9.920e-01$ |
| Accuracy | 4 | 512 | $-4.264e-01$ | $3.229e+00$ | $-9.927e-01$ |
| Loss | 2 | 64 | $-6.853e-01$ | $2.265e+00$ | $-9.975e-01$ |
| Loss | 2 | 128 | $-6.562e-01$ | $2.311e+00$ | $-9.988e-01$ |
| Loss | 2 | 256 | $-5.907e-01$ | $2.174e+00$ | $-9.968e-01$ |
| Loss | 2 | 512 | $-5.848e-01$ | $2.280e+00$ | $-9.989e-01$ |
| Loss | 3 | 64 | $-6.739e-01$ | $2.228e+00$ | $-9.980e-01$ |
| Loss | 3 | 128 | $-6.229e-01$ | $2.224e+00$ | $-9.976e-01$ |
| Loss | 3 | 256 | $-5.581e-01$ | $2.111e+00$ | $-9.942e-01$ |
| Loss | 3 | 512 | $-5.141e-01$ | $2.057e+00$ | $-9.955e-01$ |
| Loss | 4 | 64 | $-6.039e-01$ | $1.964e+00$ | $-9.980e-01$ |
| Loss | 4 | 128 | $-5.701e-01$ | $2.014e+00$ | $-9.991e-01$ |
| Loss | 4 | 256 | $-5.379e-01$ | $2.051e+00$ | $-9.951e-01$ |
| Loss | 4 | 512 | $-4.810e-01$ | $1.957e+00$ | $-9.937e-01$ |

Table 2: Details of the linear approximation of the Arxiv Dataset. Note that in this case, we used only the values of the generalization gap whose training error is below $95\%$.

| Type | Lay. | Feat. | Slope | Point | Pearson Correlation Coefficient |
|---|---|---|---|---|---|
| Accuracy | 2 | 16 | $-2.839e-01$ | $2.022e+00$ | $-9.803e-01$ |
| Accuracy | 2 | 32 | $-2.917e-01$ | $2.014e+00$ | $-9.690e-01$ |
| Accuracy | 2 | 64 | $-3.006e-01$ | $2.021e+00$ | $-9.686e-01$ |
| Accuracy | 3 | 16 | $-2.656e-01$ | $1.996e+00$ | $-9.891e-01$ |
| Accuracy | 3 | 32 | $-2.637e-01$ | $2.008e+00$ | $-9.679e-01$ |
| Accuracy | 3 | 64 | $-2.581e-01$ | $1.981e+00$ | $-9.870e-01$ |
| Loss | 2 | 16 | $-3.631e-01$ | $9.406e-01$ | $-9.250e-01$ |
| Loss | 2 | 32 | $-4.228e-01$ | $9.638e-01$ | $-9.657e-01$ |
| Loss | 2 | 64 | $-4.991e-01$ | $1.067e+00$ | $-9.776e-01$ |
| Loss | 3 | 16 | $-4.131e-01$ | $1.276e+00$ | $-9.753e-01$ |
| Loss | 3 | 32 | $-4.605e-01$ | $1.385e+00$ | $-9.730e-01$ |
| Loss | 3 | 64 | $-4.589e-01$ | $1.455e+00$ | $-9.756e-01$ |

Table 3: Details of the linear approximation of the Cora Dataset. Note that in this case we used all the values given that the training accuracy is $100\%$ for all nodes.

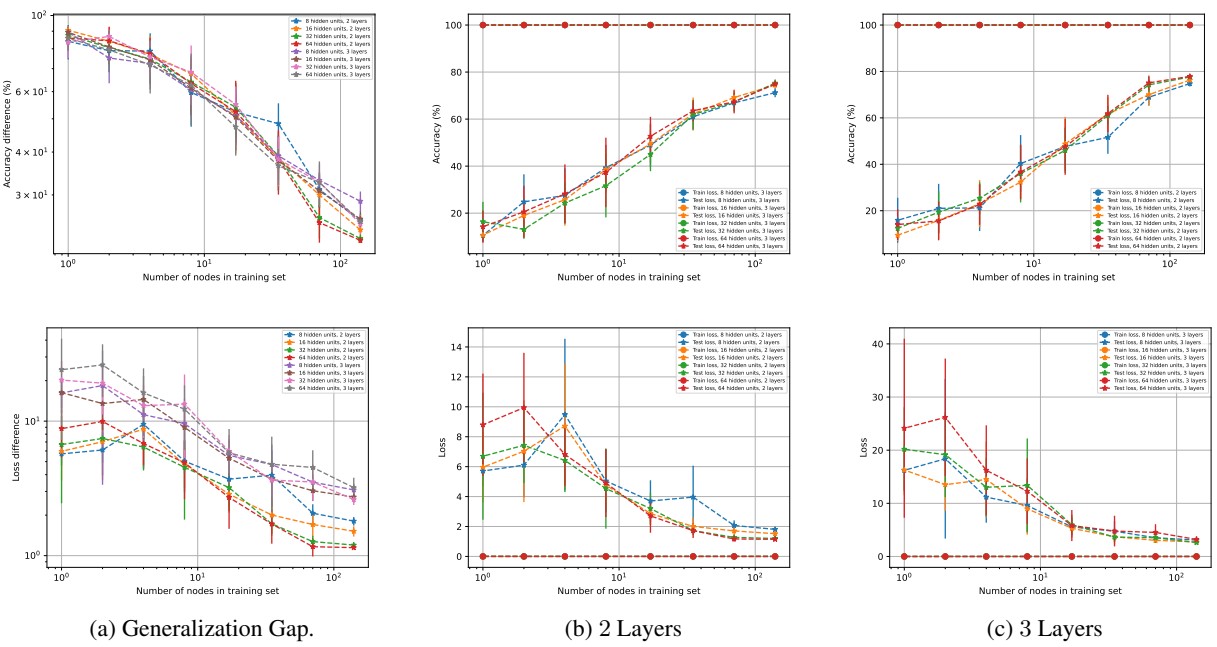

(a) Generalization Gap.          (b) 2 Layers          (c) 3 Layers

Figure 12: Generalization gap, testing, and training losses with respect to the number of nodes in the Cora dataset. The top row is in accuracy, and the bottom row is the cross-entropy loss.

### J.3.3. CiteSeer dataset

For the CiteSeer dataset, we used the standard one, which can be obtained running `torch_geometric.datasets.Planetoid(root="./data",name='CiteSeer')`.

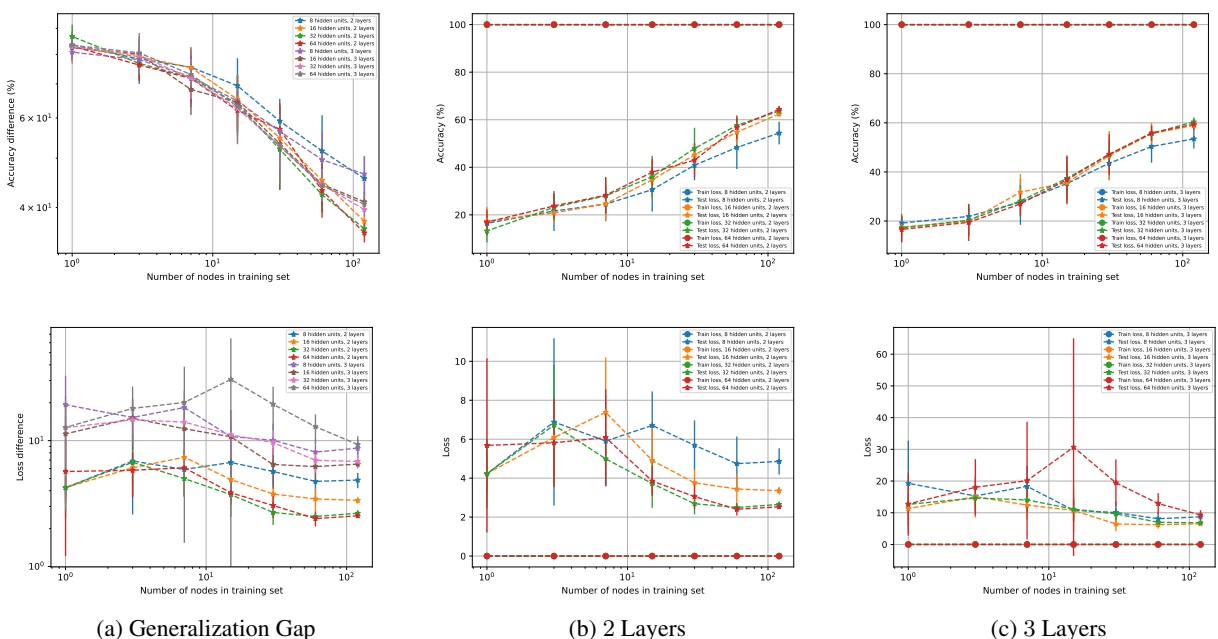

(a) Generalization Gap          (b) 2 Layers          (c) 3 Layers

Figure 13: Generalization gap, testing, and training losses with respect to the number of nodes in the CiteSeer dataset. The top row is in accuracy, and the bottom row is the cross-entropy loss.

| Type | Lay. | Feat. | Slope | Point | Pearson Correlation Coefficient |
|---|---|---|---|---|---|
| Accuracy | 2 | 16 | $-1.699e-01$ | $1.972e+00$ | $-9.518e-01$ |
| Accuracy | 2 | 32 | $-1.856e-01$ | $1.978e+00$ | $-9.714e-01$ |
| Accuracy | 2 | 64 | $-1.749e-01$ | $1.966e+00$ | $-9.534e-01$ |
| Accuracy | 3 | 16 | $-1.585e-01$ | $1.956e+00$ | $-9.721e-01$ |
| Accuracy | 3 | 32 | $-1.659e-01$ | $1.963e+00$ | $-9.721e-01$ |
| Accuracy | 3 | 64 | $-1.658e-01$ | $1.967e+00$ | $-9.702e-01$ |
| Loss | 2 | 16 | $-1.049e-01$ | $7.757e-01$ | $-5.924e-01$ |
| Loss | 2 | 32 | $-1.762e-01$ | $7.646e-01$ | $-7.981e-01$ |
| Loss | 2 | 64 | $-2.186e-01$ | $8.384e-01$ | $-9.120e-01$ |
| Loss | 3 | 16 | $-1.802e-01$ | $1.169e+00$ | $-8.345e-01$ |
| Loss | 3 | 32 | $-1.629e-01$ | $1.200e+00$ | $-8.767e-01$ |
| Loss | 3 | 64 | $-5.917e-02$ | $1.283e+00$ | $-2.562e-01$ |

Table 4: Details of the linear approximation of the CiteSeer Dataset. Note that in this case we used all the values given that the training accuracy is 100% for all nodes.

### J.3.4. PUBMED DATASET

For the PubMed dataset, we used the standard one, which can be obtained running `torch_geometric.datasets.Planetoid(root="./data",name='PubMed')`.

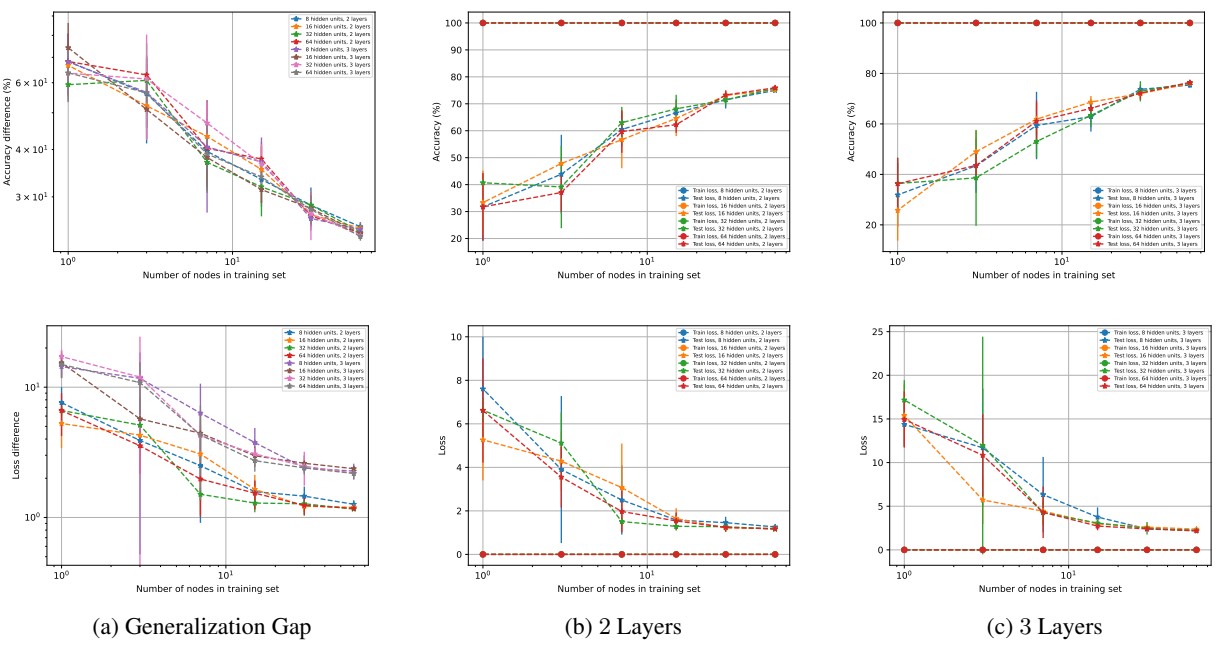

(a) Generalization Gap      (b) 2 Layers      (c) 3 Layers

Figure 14: Generalization gap, testing, and training losses with respect to the number of nodes in the PubMed dataset. The top row is in accuracy, and the bottom row is the cross-entropy loss.

### J.3.5. COAUTHORS CS DATASET

For the CS dataset, we used the standard one, which can be obtained running `torch_geometric.datasets.Coauthor(root="./data", name='CS')`. In this case, given that there are no training and testing sets, we randomly partitioned the datasets and used 90% of the samples for training and the

| Type | Lay. | Feat. | Slope | Point | Pearson Correlation Coefficient |
|---|---|---|---|---|---|
| Accuracy | 2 | 16 | $-2.523e-01$ | $1.834e+00$ | $-9.942e-01$ |
| Accuracy | 2 | 32 | $-2.433e-01$ | $1.812e+00$ | $-9.583e-01$ |
| Accuracy | 2 | 64 | $-2.764e-01$ | $1.869e+00$ | $-9.761e-01$ |
| Accuracy | 3 | 16 | $-2.748e-01$ | $1.844e+00$ | $-9.910e-01$ |
| Accuracy | 3 | 32 | $-2.661e-01$ | $1.861e+00$ | $-9.712e-01$ |
| Accuracy | 3 | 64 | $-2.558e-01$ | $1.827e+00$ | $-9.890e-01$ |
| Loss | 2 | 16 | $-4.166e-01$ | $7.695e-01$ | $-9.718e-01$ |
| Loss | 2 | 32 | $-4.733e-01$ | $7.852e-01$ | $-9.137e-01$ |
| Loss | 2 | 64 | $-4.368e-01$ | $7.547e-01$ | $-9.718e-01$ |
| Loss | 3 | 16 | $-4.424e-01$ | $1.067e+00$ | $-9.549e-01$ |
| Loss | 3 | 32 | $-5.518e-01$ | $1.223e+00$ | $-9.655e-01$ |
| Loss | 3 | 64 | $-5.246e-01$ | $1.169e+00$ | $-9.632e-01$ |

Table 5: Details of the linear approximation of the PubMed Dataset. Note that in this case we used all the values given that the training accuracy is $100\%$ for all nodes.

remaining $10\%$ for testing.

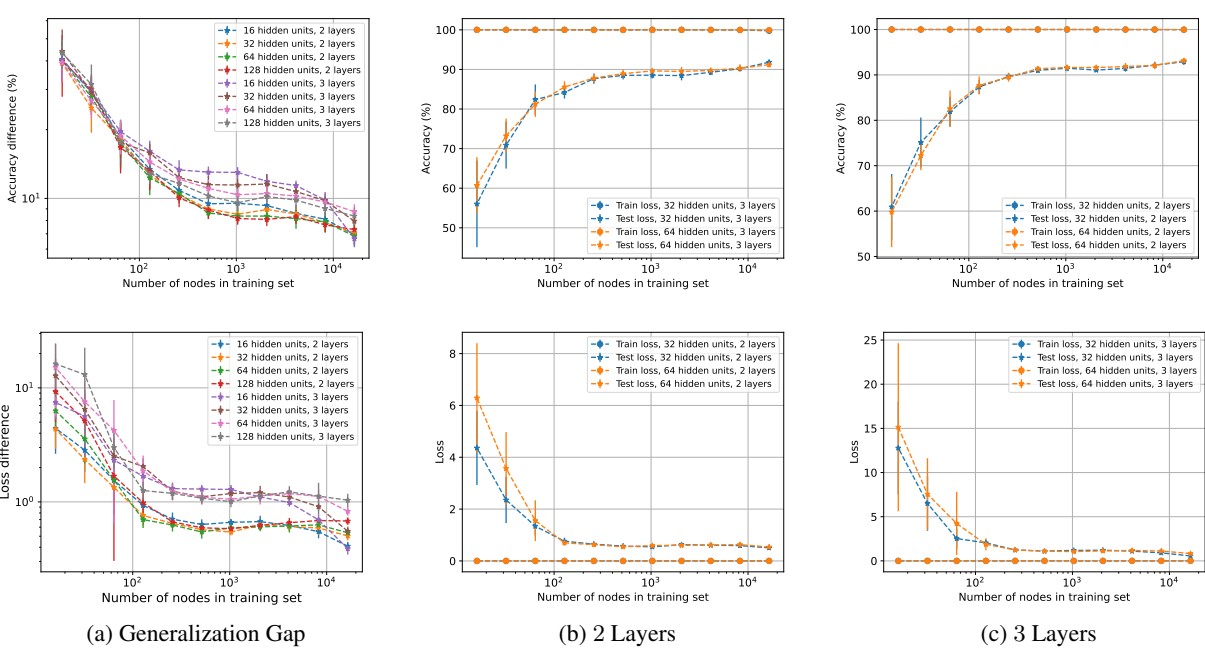

(a) Generalization Gap       (b) 2 Layers       (c) 3 Layers

Figure 15: Generalization gap, testing, and training losses with respect to the number of nodes in the CS dataset. The top row is in accuracy, and the bottom row is the cross-entropy loss.

### J.3.6. COAUTHORS PHYSICS DATASET

For the Physics dataset, we used the standard one, which can be obtained running `torch_geometric.datasets.Coauthor(root="./data", name='Physics')`. In this case, given that there are no training and testing sets, we randomly partitioned the datasets and used $90\%$ of the samples for training and the remaining $10\%$ for testing.

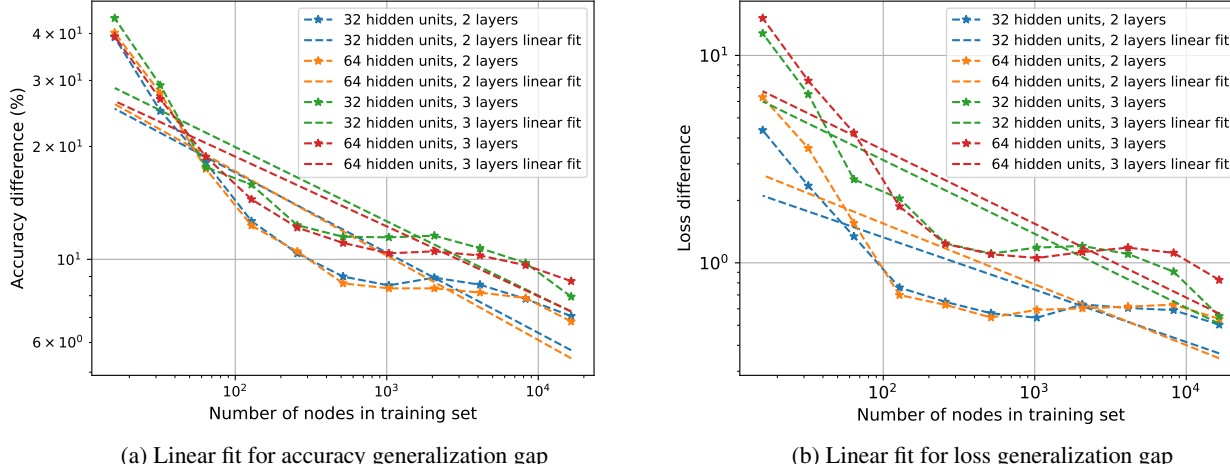

(a) Linear fit for accuracy generalization gap

(b) Linear fit for loss generalization gap

Figure 16: Generalization gaps as a function of the number of nodes in the training set in the CS dataset.

| Type | Lay. | Feat. | Slope | Point | Pearson Correlation Coefficient |
|---|---|---|---|---|---|
| Accuracy | 2 | 32 | $-2.138e-01$ | $1.659e+00$ | $-9.007e-01$ |
| Accuracy | 2 | 64 | $-2.250e-01$ | $1.685e+00$ | $-8.969e-01$ |
| Accuracy | 3 | 32 | $-1.979e-01$ | $1.695e+00$ | $-9.009e-01$ |
| Accuracy | 3 | 64 | $-1.862e-01$ | $1.646e+00$ | $-8.980e-01$ |
| Loss | 2 | 32 | $-2.523e-01$ | $6.273e-01$ | $-8.244e-01$ |
| Loss | 2 | 64 | $-2.933e-01$ | $7.762e-01$ | $-7.925e-01$ |
| Loss | 3 | 32 | $-3.558e-01$ | $1.207e+00$ | $-8.924e-01$ |
| Loss | 3 | 64 | $-3.560e-01$ | $1.256e+00$ | $-8.568e-01$ |

Table 6: Details of the linear approximation of the CS Dataset. Note that in this case we used all the values given that the training accuracy is $100\%$ for all nodes.

| Type | Lay. | Feat. | Slope | Point | Pearson Correlation Coefficient |
|---|---|---|---|---|---|
| Accuracy | 2 | 32 | $-1.524e-01$ | $1.235e+00$ | $-9.064e-01$ |
| Accuracy | 2 | 64 | $-1.478e-01$ | $1.218e+00$ | $-9.145e-01$ |
| Accuracy | 3 | 32 | $-1.227e-01$ | $1.190e+00$ | $-9.328e-01$ |
| Accuracy | 3 | 64 | $-1.268e-01$ | $1.200e+00$ | $-8.826e-01$ |
| Loss | 2 | 32 | $-1.111e-01$ | $-5.257e-02$ | $-7.591e-01$ |
| Loss | 2 | 64 | $-9.684e-02$ | $-7.335e-02$ | $-7.696e-01$ |
| Loss | 3 | 32 | $-1.410e-01$ | $2.875e-01$ | $-8.280e-01$ |
| Loss | 3 | 64 | $-1.068e-01$ | $2.388e-01$ | $-7.679e-01$ |

Table 7: Details of the linear approximation of the Physics Dataset. Note that in this case we used all the values given that the training accuracy is $100\%$ for all nodes.

J.3.7. HETEROPHILOUS AMAZON RATINGS DATASET

For the Amazon dataset, we used the standard one, which can be obtained running `torch_geometric.datasets.HeterophilousGraphDataset(root="./data", name='Amazon')`.

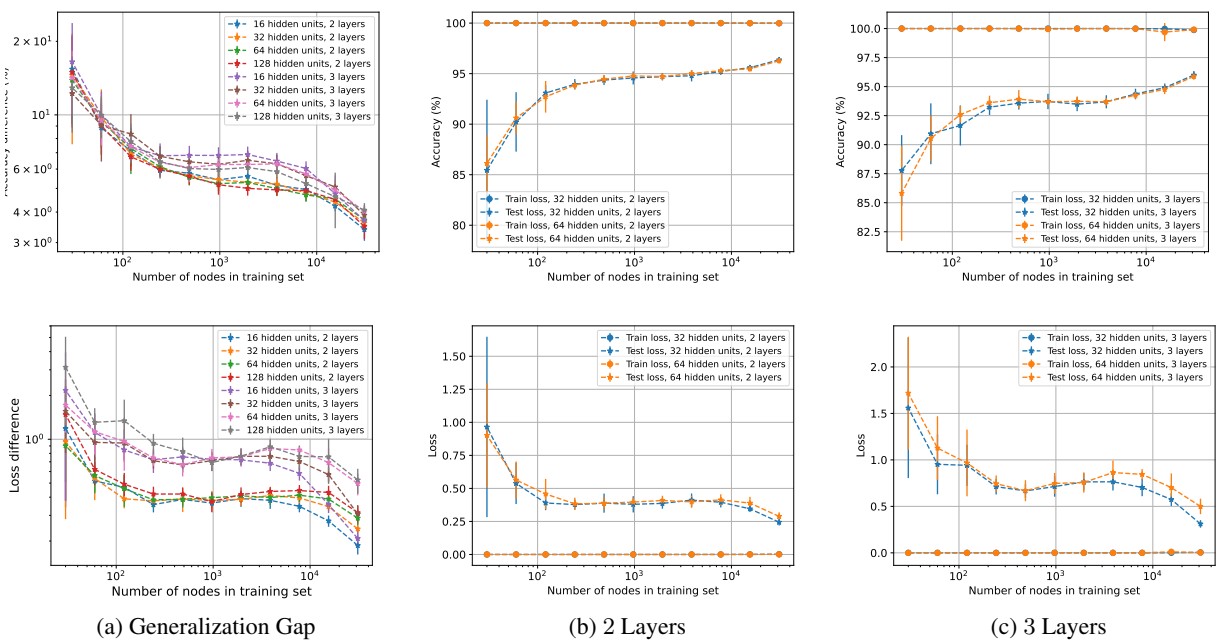

(a) Generalization Gap         (b) 2 Layers         (c) 3 Layers

Figure 17: Generalization gap, testing, and training losses with respect to the number of nodes in the Physics dataset. The top row is in accuracy, and the bottom row is the cross-entropy loss.

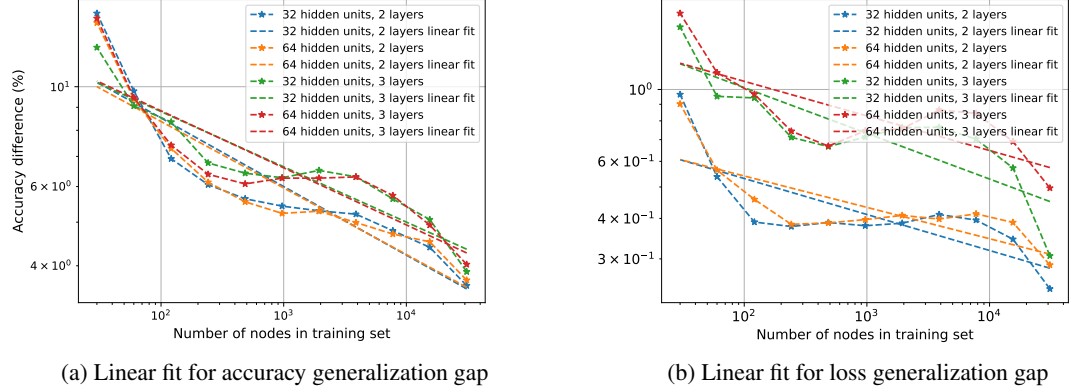

(a) Linear fit for accuracy generalization gap         (b) Linear fit for loss generalization gap

Figure 18: Generalization Gaps as a function of the number of nodes in the training set in the Physics dataset.

In this case, we used the 10 different splits that the dataset has assigned.

### J.3.8. HETEROPHILOUS ROMAN EMPIRE DATASET

For the Roman dataset, we used the standard one, which can be obtained running `torch_geometric.datasets.HeterophilousGraphDataset(root="./data", name='Roman')`. In this case, we used the 10 different splits that the dataset has assigned.

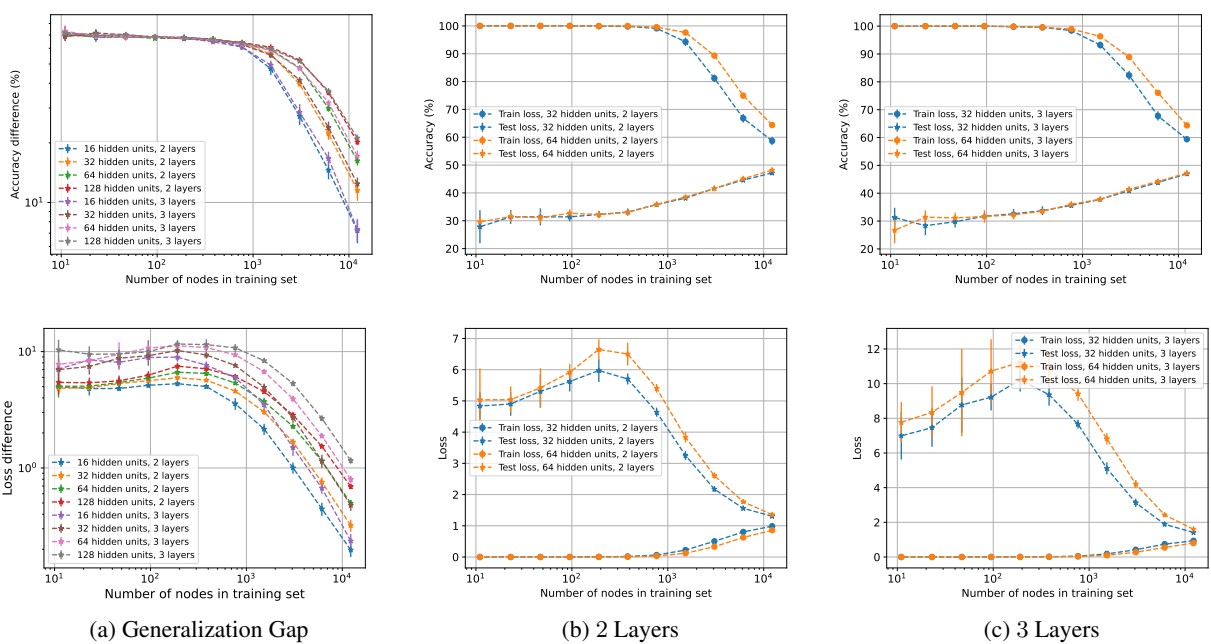

(a) Generalization Gap      (b) 2 Layers      (c) 3 Layers

Figure 19: Generalization gap, testing, and training losses with respect to the number of nodes in the Amazon dataset. The top row is in accuracy, and the bottom row is the cross-entropy loss.

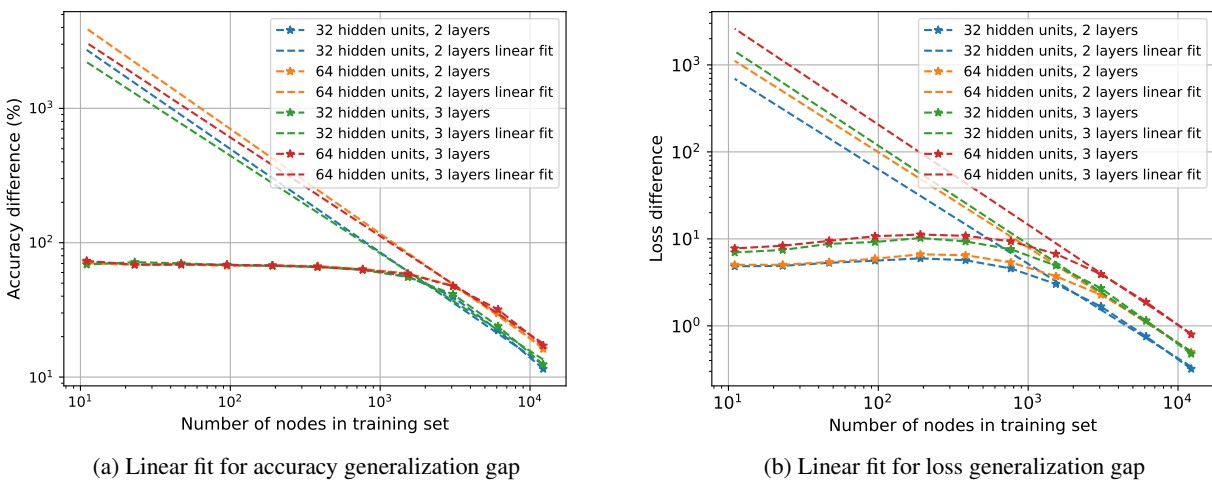

(a) Linear fit for accuracy generalization gap      (b) Linear fit for loss generalization gap

Figure 20: Generalization Gaps as a function of the number of nodes in the training set in the Amazon dataset.

| Type | Lay. | Feat. | Slope | Point | Pearson Correlation Coefficient |
|------|------|-------|-------|-------|--------------------------------|
| Accuracy | 2 | 32 | $-7.693e-01$ | $4.236e+00$ | $-9.914e-01$ |
| Accuracy | 2 | 64 | $-7.788e-01$ | $4.404e+00$ | $-9.972e-01$ |
| Accuracy | 3 | 32 | $-7.268e-01$ | $4.101e+00$ | $-9.868e-01$ |
| Accuracy | 3 | 64 | $-7.354e-01$ | $4.257e+00$ | $-9.921e-01$ |
| Loss | 2 | 32 | $-1.086e+00$ | $3.971e+00$ | $-9.968e-01$ |
| Loss | 2 | 64 | $-1.096e+00$ | $4.189e+00$ | $-9.985e-01$ |
| Loss | 3 | 32 | $-1.134e+00$ | $4.339e+00$ | $-9.965e-01$ |
| Loss | 3 | 64 | $-1.154e+00$ | $4.629e+00$ | $-9.991e-01$ |

Table 8: Details of the linear approximation of the Amazon Dataset. Note that in this case we used only the values of the generalization gap whose training error is below $95\%$.

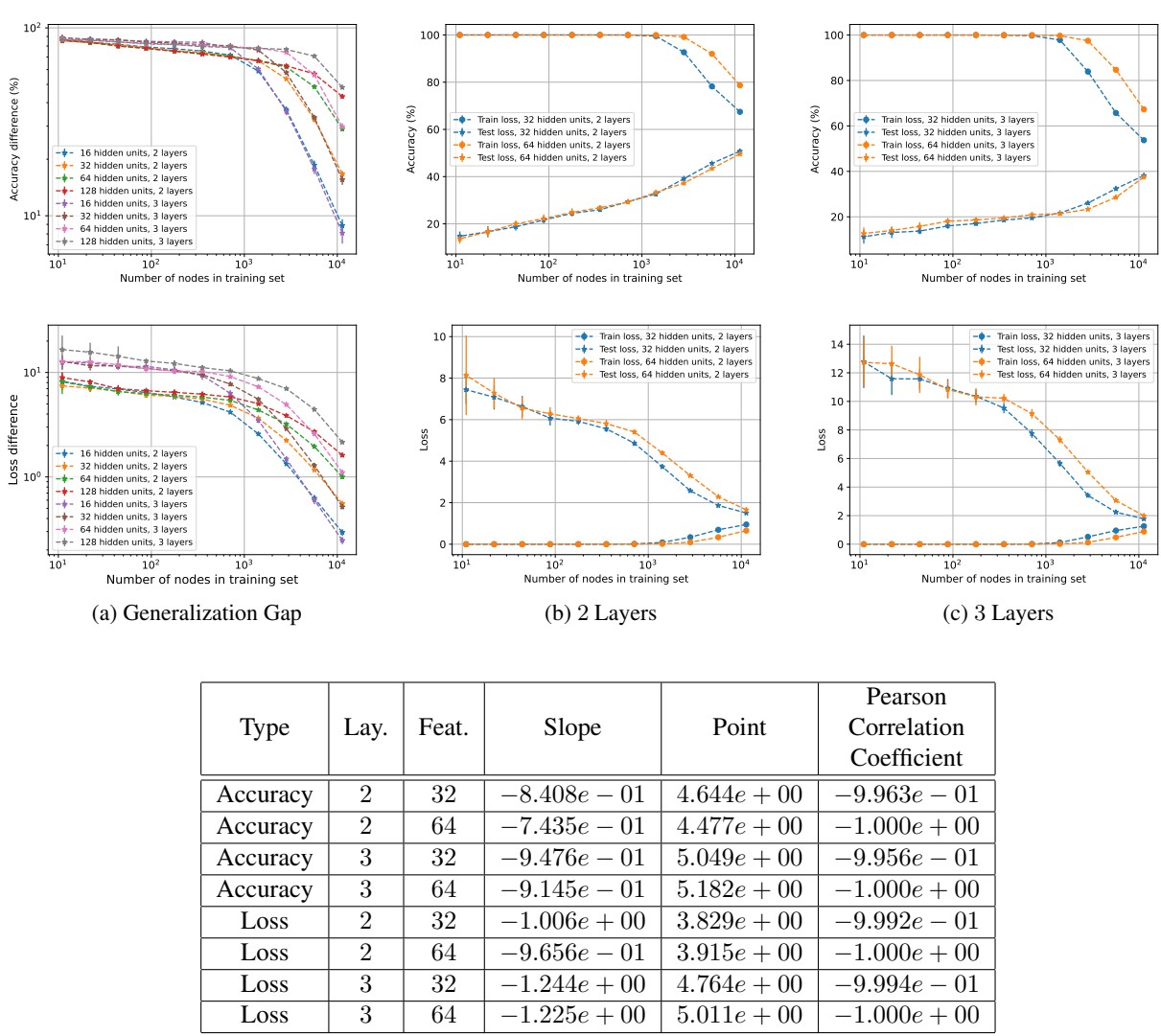

(a) Generalization Gap    (b) 2 Layers    (c) 3 Layers

| Type | Lay. | Feat. | Slope | Point | Pearson Correlation Coefficient |
|------|------|-------|-------|-------|--------------------------------|
| Accuracy | 2 | 32 | $-8.408e-01$ | $4.644e+00$ | $-9.963e-01$ |
| Accuracy | 2 | 64 | $-7.435e-01$ | $4.477e+00$ | $-1.000e+00$ |
| Accuracy | 3 | 32 | $-9.476e-01$ | $5.049e+00$ | $-9.956e-01$ |
| Accuracy | 3 | 64 | $-9.145e-01$ | $5.182e+00$ | $-1.000e+00$ |
| Loss | 2 | 32 | $-1.006e+00$ | $3.829e+00$ | $-9.992e-01$ |
| Loss | 2 | 64 | $-9.656e-01$ | $3.915e+00$ | $-1.000e+00$ |
| Loss | 3 | 32 | $-1.244e+00$ | $4.764e+00$ | $-9.994e-01$ |
| Loss | 3 | 64 | $-1.225e+00$ | $5.011e+00$ | $-1.000e+00$ |

Table 9: Details of the linear approximation of the Roman Dataset. Note that in this case we used only the values of the generalization gap whose training error is below $95\%$

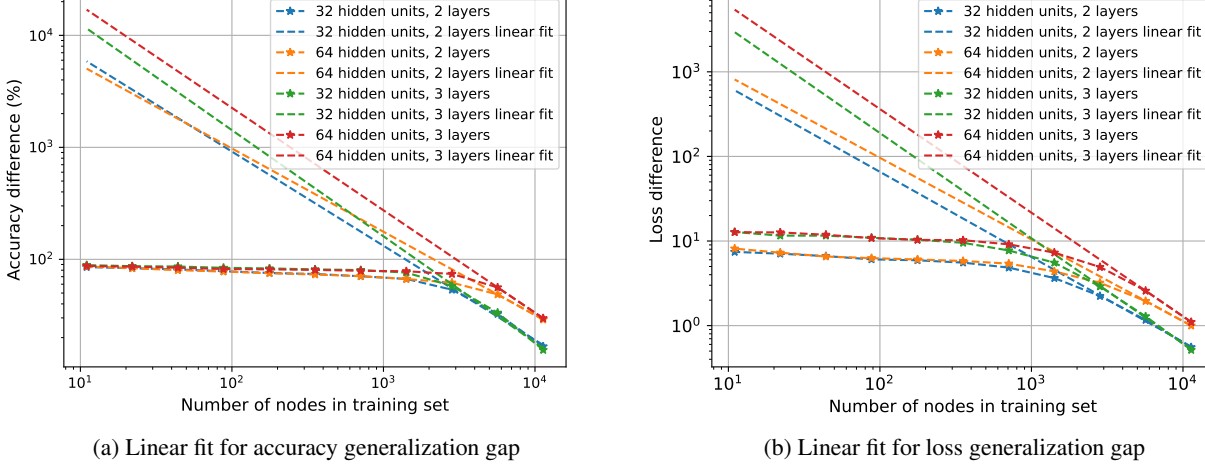

(a) Linear fit for accuracy generalization gap

(b) Linear fit for loss generalization gap

Figure 22: Generalization Gaps as a function of the number of nodes in the training set in the Roman dataset.

