# OpenReview forum: "A Manifold Perspective on the Statistical Generalization of Graph Neural Networks"
_ICML.cc/2025/Conference — ICML 2025 poster_

### Official Review · Reviewer_k2or · 2025-03-08

**Overall Recommendation:** 2

**Summary:**

The paper addresses the question of generalization in GNNs when the graph is a discrete sample from the graph. They prove this theoretically, as well as experiment with several existing datasets.

**Claims And Evidence:**

I have an issue with the empirical evidence. The main thing one can see is that the gap between the training and test loss decreases with the number of training examples. This is not very surprising or insightful, and as the real graphs are not generated in a way that is congruent with the theory, I am not sure how do they relate to the theoretical part.

**Essential References Not Discussed:**

NA

**Experimental Designs Or Analyses:**

Discussed previously

**Methods And Evaluation Criteria:**

As stated before, not sure how the experiment support the theoretical part.

**Other Comments Or Suggestions:**

Small remark- Fig. 2&3 are very unclear and should be improved or removed.

**Other Strengths And Weaknesses:**

I have a few concerns regarding this paper:
- I am not sure about the novel contribution of this paper w.r.t Weng et al "Geometric Graph Filters and Neural Networks:
Limit Properties and Discriminability Trade-offs"
-  I wouldn't say the results are really about generalization, this is more approximation results that show how the function on the manifold can be approximated with a finite sample. The graph level results only shows results on graphs on the seen manifolds, and no results on generalization to new manifolds. I do not think the way the authors present their results is aligned with what they actually show. S

**Questions For Authors:**

NA

**Relation To Broader Scientific Literature:**

The claims in the paper sem very similar to theorem 2&3 from Weng et al "Geometric Graph Filters and Neural Networks:
Limit Properties and Discriminability Trade-offs". Please state the exact difference between these results.

**Theoretical Claims:**

I did check the correctness, but I did not have time to comb over every part of the proofs. To the best of my understanding the proofs are ok.

---

> ### Author Rebuttal · Authors · 2025-04-01
>
> >**The main thing one can see is that the gap between the training and test loss decreases with the number of training examples. This is not very surprising or insightful, and as the real graphs are not generated in a way that is congruent with the theory, I am not sure how do they relate to the theoretical part.**
>
> The reviewer states that the decrease in the generalization gap with respect to the training examples is not very surprising or insightful. This is an empirical observation, and what our paper provides is a theoretical explanation of why this can be the case. That is to say, given that the decrease happens in practice, we aim to explain it. To do so, we leverage a geometric model for graphs that is insightful and interpretable -- the manifold.
> In terms of novelty, let us point out 3 aspects that to the best of our knowledge are novel and insightful about our work:
> 1. Our conclusions are not only that there is a decrease in the gap as a function of the number of nodes, but rather that **the rate of decrease** is consistent with what our theory predicts. This experiment, to the best of our knowledge, is novel in the literature for real world graphs. *Z: We further introduce the spectral continuity and reveals its relationship with the generalization ability. The provides a novel complexity measure over GNN models.*
> 2. A second novelty of our work relies on the fact that there is a unifying theory that explains both node prediction (Theorem 1) as well as graph prediction (Theorem 2). To the best of our theory, our work is the one that allows both problems to be explained.
> 3. Our theory relies on manifolds, which are a more intuitive model that better captures the geometry of the data. In Appendix I, Figure 7, we plot the spectral decay in the graph eigenvalues for 8 datasets. As can be seen, in the Figure, the sharp decay in the eigenvalues aligns with the underlying manifold assumption.
>
> >**I am not sure about the novel contribution of this paper w.r.t Weng et al "Geometric Graph Filters and Neural Networks: Limit Properties and Discriminability Trade-offs"**
>
> Our work and [1] have important differences.
> 1. In terms of the problem, our paper looks into a different problem than the one in [1]. In the case of [1] there is no performance measurements (i.e. loss functions) of machine learning involved. That is to say, the problem [1] tackles is the output function approximation of GNNs, and not the generalization ability of GNNs.
> 2. In terms of theory, our results differ from [1]. Instead of providing a probability bound depending on a sampled graph from the manifold as in [1], we derive a  uniform bound over the space of functions of the sampled graphs, which is akin to the setting of machine learning generalization analysis.
> 3. In terms of architecture, in [1] they consider graph input/graph output architectures. In our case, we consider both node level classification as well as graph level classification problems.
> In all, although related, our paper is novel with respect to [1].
>
> [1] "Geometric Graph Filters and Neural Networks: Limit Properties and Discriminability Trade-offs", Zhiyang Wang, Luana Ruiz, Alejandro Ribeiro
>
> >**I wouldn't say the results are really about generalization, this is more approximation results that show how the function on the manifold can be approximated with a finite sample.**
>
> In our work, we interpret the distribution of graph nodes as lying on an underlying manifold. This perspective aligns with the standard assumption in machine learning, where the data points (in our case, nodes) are sampled from a certain distribution (in our case, manifold). Unlike purely abstract or discrete distributions, our manifold-based interpretation explicitly captures and leverages geometric and structural relationships intrinsic to the nodes and their interactions in the graph.
> With this perspective, the statistical risk — the expected loss of our neural network — is naturally expressed as the integral or average of the loss over this node manifold. The work in "Geometric Graph Filters and Neural Networks: Limit Properties and Discriminability Trade-offs" focus on the output difference analysis over the sampled points from the manifold, ignoring the unseen or unsampled points over the manifold. Our work complements this by providing this statistical analysis view.
> The problem that we are considering is what it is called generalization in machine learning, see for example:
> [Chapter 3] Understanding Machine Learning: From Theory to Algorithms
> By Shai Shalev-Shwartz and Shai Ben-David
> Cambridge University Press;
> [Chapter 2] Foundations of Machine Learning
> Mehryar Mohri, Afshin Rostamizadeh, and Ameet Talwalkar
> MIT Press, Second Edition, 2018. https://cs.nyu.edu/~mohri/mlbook/;
> [Equations 1,2,3] Generalization analysis of message-passing neural networks on large random graphs
> S Maskey*, R Levie*, Y Lee, G Kutyniok
> NeurIPS 2022

---

### Official Review · Reviewer_EGQX · 2025-03-12

**Overall Recommendation:** 4

**Summary:**

This paper considers GNNs where on graphs which arise from subsampling a manifold (with non-uniform density), building off of a number of recent works which have analyzed the convergence of such networks.

However, this paper adds an exciting new dimension to this line of works by incorporating ideas from statistical learning theory to analyze the generalization gap. As a practical take away, Corollary 1 provides transfer learning guarantees between two graphs G_1 and G_2 sampled from the same manifold.

Currently, there is a significant number of minor errors in the proof. I believe that all of them are easily fixable, but some require minor modifications to the results. (Notably, I think that the statement of the theorem might not apply when $d=1$ and may need minor changes in the case $d=3). These are detailed below.

I think that this paper makes a significant contribution and am recommending acceptance. However, since these contributions are primarily theoretical, it is CRITICAL that the proofs be 100% correct. Therefore, I am noting the at my positive score is conditioned on the assumption that all of errors noted below be fixed in the camera copy. (Except for in any cases where I am incorrect, which should be discussed in reviewer discussion.) All of the errors are easily fixable, so I do not anticipate the authors having difficulty with them.

**Claims And Evidence:**

yes

**Essential References Not Discussed:**

First paper (to my knowledge) to provide convergence rates for LBO derived deep networks as the number of samples -> \infty
https://www.sciencedirect.com/science/article/pii/S1063520324000125

Extension of the previous work to more general deep networks. (To the best of my knowledge) this was the first paper to do this with a quantitative convergence rate (although Wang 2024a established a similar result without a rate) https://ieeexplore.ieee.org/abstract/document/10301407

Work extending MNN guarantees to non-uniform sampling which shows that the convergence rate can be improved via proper normalization and also applies to k-NN graphs (with a different weighted manifold Laplacian) https://arxiv.org/abs/2307.04056

**Experimental Designs Or Analyses:**

yes

**Methods And Evaluation Criteria:**

yes

**Other Comments Or Suggestions:**

N/A

**Other Strengths And Weaknesses:**

3.1 In Setup – should specify that D and A are weighted degree matrices and weighted adjacency matrices
Should also explain why a graph signal is a vector. (This is a slight abuse of notation since it is nominally a function. I understand that this is “standard” in GSP, but I think it is best to clarify that you are equating the function x:VR with the vector x_i = x(v_i).
3.2 – Unclear what is meant by a ``Hausdorff Probability Measure.” The term Hausdorff measure usually refers to fractal (non-integer) dimensions. I think you mean that the measure is absolutely continuous with respect to the Riemannian  volume for
4.1 - The assumption A.3 that the loss function is Lipschitz continuous doesn’t hold for common loss-functions such as the MSE or cross-entropy. This is okay, but should be properly discussed, especially since the cross-entropy loss was used in your experiments
5. The current experiments are good, but it would be better to include some graphs that are clearly derived from a manifold. I understand that the authors show that there is a rapid decay of the graph Laplacian, but this does not directly indicate that the graph is a subsample of a manifold. Weyl’s law says that if the graph is from a manifold, then the Laplacian eigenspectrum will decay rapidly, but there is not a converse to this. (The modelnet experiments are good, but 2-d surfaces in 3-d space are fairly non-generic manifolds, which may not have co-dimension 1. Additionally, those shapes are likely not true manifolds because of corners.)

It is useful that the authors show that their theory applys to non-manifold graphs, but a direct illustration of the theory would also be beneficial. Perhaps the authors could look at the experimental setup in Johnson et al. https://arxiv.org/abs/2307.04056 where graphs are constructed via subsampling an ellipsoid (embedded in high-dimensional space).

Appendix A: should clarify whether B_r is a ball with respect to the Euclidean distance (in ambient space) or the geodesic distance.

Appendix A: VERY IMPORTANT – looking at Garcia Trillos et al, it appears that the bound on r only applies to d >= 3. There is a different result for d=2 and no result for d=1. Please update the theorem of your statement accordingly.

Appendix A: VERY IMPORTANT The bounds in Prop 1 appear to depend on the bandlimit M. Please update the statement of the theorem (as well as Theorem 1) to reflect this.

Appendix A: VERY IMPORTANT: The result you recall from Wang 2024a appears to be off (see (20) and (23) of Wang2024a) it looks like the result there features a square root epsilon and doesn’t have the lambda_i. Pleas also update all downstream results accordingly. Less importantly, Prop 4 of Wang 2024a appears to be a restatement of Calder and Trillos. Please make this more clear.

Equation (36): shouldn’t |h’(\lambda_i)| be replaced with \sup_{lambda in [lambda_i,N – lamda_i| h’(\lambda)? (This won’t affect the next step, but should still be fixed)

Equation 42: I get lambda_i^{-d+1} rather than lambda_i^{-d}. Since there is a lambda^{-d} in the estimate of h and an lambda in your bound on ||phi_i,N – P_N phi_i||. Please double check this calculation. Notably, this means that the series will no longer be summable from 1 to infinity if d=2, but this is okay since it is a finite sum and A_2(M,N) already depends on the bandlimit

Should recall Weyls law after the statement of 58 in order to make life easier for the reader

Proposition 3, why is there a C in the statement but a C’ in (71)?

It appears that the Lipschitz constant C depends on the bandlimit M.  This should be made clear in the statement of the proposition

I don’t see how (74) follows from 72. It seems like there should be various sup’s there. I am confident that the result is true, but I think a simple induction proof would be much better. This should be fixed for the camera copy. (I think it this case, you can prove the result for L = 1 and then say that the general case follows by induction.)

Prop 3: The statement of the proposition should make clear that you are assuming f is bandlimited. I understand you make this assumption in theorem 1, but someone might read Prop 3 independently. (Alternatively, a good way to avoid this is to start Prop 3 with “assume the assumptions of theorem 1 hold”)

Prop 3: What is B_r(M)? I think you mean y \in B_r(x)? (B_r(M) could mean the points in the ambient space which are close to the manifold)

In the Proof of Theorem 1, it is incorrect to call the V_i “Voronoi cells”, if you had actually used the Voronoi decomposition, as opposed the the V_i induced by the OT map, then the V_i would not all have the same measure.

Minor
Some notational inconsistency with subscripts. For example, in equation (1) there is a subscript G, but there is no subscript M in equation (4).

There are some uncapitalized words like Laplacian and Euclidean in the references. Please double check your bibtex entries

Appendix A: To help the reader, it would be good to note that P_NI_Nx = x but that the we don’t have I_NP_N f = f. (This is a common source of confusion.)

Appendix A: Line 862 “equation equation” (please check for this throughout)

**Questions For Authors:**

It appears that you are assuming a single input channel. Is this restriction necessary? It appears that most of the theory can be extended to multiple input channels. I don’t think you should re-do the analysis, but maybe make add a remark to increase the impact of your theory

Similarly, in theorem 1, it appears you could remove the assumption that there is a constant number of filters per layer. Is this correct? If so, this could be a good thing to comment on briefly

Should there be a 1/K in the definition of the risk functions in (17) and (18)? This would seem to be consistent with the 1/N_k

Plotting the decay of the eigenvalues is interesting. Would it be possible to use this infer the manifold dimension via Weyl’s law via plotting the eigenvalues on a log plot?

**Relation To Broader Scientific Literature:**

Good discussion of GNNs and an okay review of manifold neural networks and manifold learning with some missing references noted below

**Theoretical Claims:**

Proofs were thoroughly checked. Mostly correct. However, there are a large number of (easily fixable errors) as detailed below. My score is predicated on the assumption that the authors will fix these mistakes in the camera copy.

---

> ### Author Rebuttal · Authors · 2025-04-01
>
> We thank the reviewer for giving us a thorough check of our proof and all the suggestions. We are glad to find the reviewer think our work as ''exciting'' and ''significant''. We have carefully considered and addressed all the minor concerns that the reviewer has pointed out and we will make the accordant updates to guarantee that our results are 100% correct. We are addressing some of the questions to further clarify our points.
> - In Section 3.2, the reviewer is correct that we are assuming the measure is absolutely continuous with respect to the Riemannian volume. We will change the ''Hausdorff probability measure'' to a non-vanishing Lipschitz continuous density $\rho$ with respect to the Riemannian volume on $\mathcal{M}$.
> - The reviewer is correct that some common loss functions such as cross-entropy loss that we used is not strictly globally Lipschitz continuous. While it can be locally Lipschitz continuous if restricted to subsets where probabilities are bounded strictly away from 0 and 1. This can be realized with activation functions like sigmoid that naturally produce outputs away from 0 or 1. We will add this discussion to the main context. We thank the reviewer for pointing out this important point. We will add these notes following Assumption 3.
> - We totally agree with the reviewer that it would be better to include some graphs that are clearly derived from a manifold. We have the synthetic manifold examples shown in Figure 1 to show the results on graphs derived from a practical manifold. We will implement the subsampling from an ellipsoid in [3] to apply to non-manifold graphs to make our work more general.
> - $B_r$ is a ball with respect to the Euclidean distance in the Euclidean ambient space, while $B_r(\mathcal{M})$ is defined as a ball in $\mathcal{M}$ with respect to geodesic distance on $\mathcal{M}$. We will add these explanations in the updated version.
> - We thank the reviewer for pointing out that the bound on $r$ only applies to $d \geq 3$. We will add the case for $d=2$ separately.
> - The bounds in Prop 1 appear to depend on the band limit $M$ -- We suppose $M$ to be large enough such that $M^{-1}\leq \delta'$ and $C_2$ is related to $\delta'$ as we stated in line 967. We will make more elaborations on this.
> - We will add the statement that Prop 4 of Wang 2024a as a restatement of Calder and Trillos, and we will update the $\epsilon$ term. For the $\lambda_i$ terms, we are replacing the upper bound $\lambda_K$ in Prop 4 of Wang 2024a as we are now assuming a bandlimited scenario and can have a point-wise analysis.
> - The reviewer is right that in equation (42), we need to discuss about the case $d=2$ separately. We will add this explanation after equation (44) that the constant is different for the case $d=2$.
> - For Proposition 3, the upper bound of the norm of gradient does not need to be the same with the Lipshitz constant of $g$. While we could assume it the same for the ease of presentation. We will update this and we thank the reviewer for pointing this out.
> - For how (74) follows from (72), we are iteratively repeating the process for each layer and this would finally reduce to the single dimension input of the GNN. We will add explanations in the updated version.
> It is not necessary to assume a single input channel, but it is for ease of presentation. We will add a remark to state this potential expansion.
> - The assumption that there is a constant number of filters per layer is also for ease of presentation and we will add a remark to this as well. We thank the reviewer for helping us improve our work.
> - We agree that adding a $1/K$ in the definition of the risk functions in (17) and (18) can help with the normalization of the definition and does not have impact on the proof process.
> - It would be interesting to explore the possibility of using the eigenvalue plot to infer the manifold intrinsic dimension. This would help to alleviate the restriction of the prior manifold information.

---

> > ### Comment · Reviewer_EGQX · 2025-04-01
> >
> > I thank the authors for thoroughly addressing the (minor) concerns raised in my review

---

### Official Review · Reviewer_3hav · 2025-03-14

**Overall Recommendation:** 2

**Summary:**

This paper provides a new perspective the analyze the generalisation of GNN from manifold and manifold neural networks. By considering a graph as samples from a manifold, this paper shows that the generalisation ability of GNNs decrease with number of nodes and increases with the spectral contiuity constant (an indicator of model complexity).

**Claims And Evidence:**

The claims are mostly supported.
However, I find the claim about GNN discriminability somewhat hand-wavy. The authors claim this based on spectral contiuity which is a indicator for spectral filter complexity. But complexity and discriminability are not equal, e.g. a model can perfectly discriminate all data points but have low complexity such as linear classifier, and a model can have high complexity but still cannot discriminate data points. To properly claim this, I feel the authors should define discriminability first to make the claim rigorous.

Also, I noted the definition of generalisation gap (eq 15) is slightly different to other literature.

**Essential References Not Discussed:**

[1][2][3] should be discussed. The two papaers are very recent so it is understandable that the authors may have missed them by the time of submission.

[3] Generalization, Expressivity, and Universality of Graph Neural Networks on Attributed Graphs. ICLR 2025

**Experimental Designs Or Analyses:**

Using a regulariser to indicate spectral contiuity feels indirect and there are many uncontrolled factors, e.g. whether the regulariser works or not depends on many factors such as training settings and the weight coefficient.

**Methods And Evaluation Criteria:**

Yes. The methods are evaluated on both synthetic and real-world datasets.

**Other Comments Or Suggestions:**

N/A

**Other Strengths And Weaknesses:**

Strengths:
* The analysis works for both graph and node level tasks.
* Manifold perspective is interesting.

Other weaknesses:
* As  $C_1$, $C_2$ and $C_3$ depends on the geometry of the manifold, they are potentially important and might be a good contribution so it is a pity that the authors didn't discussed them in depth.
* The introduction of convolution and spectral filters is a bit convoluted and can be simplified.
* The spectral contiuity term can be impractical to compute.

**Questions For Authors:**

* can you please briefly describe  $C_1$, $C_2$ and $C_3$ and their implication?
* why do you use a regulariser to indicate spectral contiuity. It seems very indirect.
*

**Relation To Broader Scientific Literature:**

My main concern is the novelty of this paper. While analysing generalisation from manifold neural network is novel, the landed results are not so. In particular, the conclusion that generalisation decreases with spectral contiuity (model complexity) is well-known traditional wisdom in statistically machine learning and is known that it doesn't reflect reality. Many models of high-complexity can achieve good generalisation. In the area of GNN, this is also known that a more complex GNN can sometimes achieve better generalisation [1][2]. In this regards, this paper doesn't provide much valuable insight.
Also, the results regarding size of graph is known from (Maskey et al., 2022; 2024; Levie, 2024).

[1] Weisfeiler–Leman at the margin: When more expressivity matters. ICML 2024
[2] Towards bridging generalization and expressivity of graph neural networks . ICLR 2025

**Theoretical Claims:**

The theoretical claims seems correct. However, I am not able to follow some proofs so I could be missing things.
For example, I cannot find definition of $C_1$, $C_2$ and $C_3$. I checked the appendix but can only find $C_1$ and $C_2$ depend on $C_{\mathcal{M},1}$ and $C_{\mathcal{M},2}$, but could not find what $C_{\mathcal{M},1}$ and $C_{\mathcal{M},2}$ are.

---

> ### Author Rebuttal · Authors · 2025-04-01
>
> >**As C1, C2 and C3 depend on the geometry of the manifold, they are potentially important and might be a good contribution so it is a pity that the authors didn't discussed them in depth.**
>
> The parameters are related to the geometry of the manifold. We thought it distracting to expand in the main context. We totally agree with the reviewer that this would be a good contribution to discuss. We will add the details in the appendix to address these impacts.
>
> >**The spectral contiuity term can be impractical to compute. why do you use a regulariser to indicate spectral contiuity. It seems very indirect.**
>
> The reviewer is right that calculating the spectral continuity constant might be computationally expensive. However, approximations to it might allow an efficient implementation.
> This are for example:
> [1] Stability properties of graph neural networks
> F Gama, J Bruna, A Ribeiro
> IEEE Transactions on Signal Processing 68, 5680-5695
> [2] Yinan Huang, William Lu, Joshua Robinson, Yu Yang, Muhan Zhang, Stefanie Jegelka, Pan Li, ON THE STABILITY OF EXPRESSIVE POSITIONAL ENCODINGS FOR GRAPHS, ICLR 2024
>
> >**My main concern is the novelty of this paper. -- [1][2][3] should be discussed. this is also known that a more complex GNN can sometimes achieve better generalisation [1][2]. In this regards, this paper doesn't provide much valuable insight. Also, the results regarding size of graph is known from (Maskey et al., 2022; 2024; Levie, 2024).**
>
> We will add these latest references for discussion in our updated version. We thank the reviewers for the suggestions.
> We propose to use spectral continuity to explain generalization when regular bounds indicate that we do not generalize. Our proposed approach leverages spectral continuity, shifting the notion of complexity from traditional parameter-based measures to a spectral domain, specifically considering filters with an infinite number of coefficients. Despite the infinite-dimensional nature of these filters, our spectral-based complexity measure still leads to meaningful, empirically validated bounds. This is consistent with the empirical observations the reviewer makes that high complexity may not lead to worse generalization. Our novelty lies in that we derived the generalization bounds based on spectral complexity measures. Furthermore, the bounds we have shown and proved match the empirical evidence in our plots.
>
> In other words, our work suggests that complexity measured through a spectral lens captures essential aspects that traditional complexity measures overlook. This offers a deeper understanding of why specific complex GNN models generalize effectively.
>
>
> >**In particular, the conclusion that generalisation decreases with spectral continuity (model complexity) is well-known traditional wisdom in statistical machine learning and is known that it doesn't reflect reality. Many models of high-complexity can achieve good generalisation. In the area of GNN, this is also known that a more complex GNN can sometimes achieve better generalisation [1][2]. In this regards, this paper doesn't provide much valuable insight.**
>
> The value of our work relies on our characterizing what 'model complexity' means in the context of GNNs. Model complexity measures for neural networks are, for example, VC-dimension and Rademacher complexity. However, these two well-studied measures do not readily apply to GNNs, given that they do not leverage the underlying geometric structure of the data. Therefore, the value of our work relies precisely on identifying the role of the spectrum of the graph (or manifold) in the model.
> We agree with the reviewer that our work is related to (Maskey et al., 2022; 2024; Levie, 2024). And we believe there is value in our work, given that our bounds depend on geometric measures that are more interpretable and understandable. Also, unlike (Maskey et al., 2022; 2024; Levie, 2024), we provide a bound for node classification, which was not offered before.

---

### Official Review · Reviewer_r3TG · 2025-03-16

**Overall Recommendation:** 3

**Summary:**

This paper examines the generalizability of Graph Neural Networks (GNNs) from a manifold perspective. Leveraging spectral analysis, the authors introduce a novel generalization bound for GNNs, demonstrating that when trained on graphs sampled from a manifold, the generalization error decreases logarithmically with the number of graph nodes and is proportional to the spectral continuity constant of the graph filter. Experimental results on multiple real-world datasets (e.g., ArXiv, Citeseer) validate the theoretical findings.

**Claims And Evidence:**

Most of the claims are supported by proof or numerical evidence.

**Essential References Not Discussed:**

N/A

**Experimental Designs Or Analyses:**

Overall, the experimental designs seem sound to me.

**Methods And Evaluation Criteria:**

The authors considered 10 different datasets and two common metrics for numerical evaluation, making the results generally convincing.

**Other Comments Or Suggestions:**

Figure 2 may be incorrect, as the Gaussian kernel graph should be fully connected, but the figure does not reflect this.

Additionally, the font embedded in the figure is too small and should be adjusted for better readability.

One point I missed regarding Figure 5(a) is why the accuracy decreases as the number of nodes in the training set increases. Further explanation of this trend would be useful.

**Other Strengths And Weaknesses:**

I find the paper is a little bit difficult to follow, as many definitions and mathematical symbols require multiple steps to trace. For example, the chain of C1 spans Theorem 1 → Appendix D → Proposition 1. Additionally, the definition of the generalization gap (as well as discriminability) does not appear until page 5, despite being referenced multiple times earlier without citation. The authors should consider reminding readers where key definitions can be found.

The authors mention the trade-off between generalizability and discriminability, suggesting that restrictions should be imposed on the continuity of the filter functions. However, it is unclear how this restriction should be implemented—whether it should be based on dataset characteristics, selected model architectures, or other factors. Further clarification on this would be helpful.

**Questions For Authors:**

Please address the following points of confusion:

(1) Does non-uniform sampling matter?

(2) What concrete suggestions can be made to impose restrictions on the continuity of filter functions?

(3) Why does accuracy decrease with more nodes (Figure 5(a))?

(4) Are the proposed bounds or the manifold perspective applicable to various types of GNNs?

**Relation To Broader Scientific Literature:**

The authors provide a comprehensive discussion and comparison with existing literature, highlighting that while traditional generalization bounds grow with graph size or node degree, the proposed bound decreases with the number of nodes, driven by the spectral properties of filter functions over the manifold. Additionally, they demonstrate that a GNN trained on a single graph from each manifold can generalize to unseen graphs from the same manifold set.

**Theoretical Claims:**

Overall, the authors provide comprehensive proofs, and to my knowledge, there are no significant flaws.

However, it seems that the authors assume that the graph is uniformly sampled from the underlying manifold, which is a strong assumption. It is unclear how deviations from uniform sampling would impact the conclusions. Some discussion may be needed.

Additionally, the authors may need to be more explicitly discussed whether the generalization bounds of all GNNs align with the manifold assumption.

---

> ### Author Rebuttal · Authors · 2025-04-01
>
> > **Does non-uniform sampling matter?**
>
> The sampling does not need to be uniform, but the points must be independently and identically distributed (i.i.d.) randomly sampled according to the measure $\mu$ over the manifold. We outline this condition at the beginning of section 3.2 -- the measure $\mu$ might not be uniform. The requirement is that the density is bounded below and above:
>
> $$ 0<\rho_{min} \leq \rho(x) \leq \rho_{max}<\infty \quad \forall x \in \mathcal{M}$$
>
>
>
> >**What concrete suggestions can be made to impose restrictions on the continuity of filter functions?**
>
> The spectral continuity could be impractical to compute accurately. Therefore, to address this, we add a penalty term to the loss function to impose this continuity constraint during the training process.
>
> >**Why does accuracy decrease with more nodes (Figure 5(a))?**
>
> The training accuracy decreases with more nodes as the graph size grows. With the neural network size fixed, we expect a worse training accuracy as we train on a larger graph. This can result from the limited GNN model experiencing underfitting over the growing graph.
>
> >**Are the proposed bounds or the manifold perspective applicable to various types of GNNs?**
>
> Yes, we prove for a general GNN convolutional form, and this can be extended to include other specific GNN convolutional models: https://pytorch-geometric.readthedocs.io/en/latest/modules/nn.html#convolutional-layers
>
> >**Additionally, the font embedded in the figure is too small and should be adjusted for better readability.**
>
> Thank you for the suggestions and other comments to help us improve the paper. We will update these.
>
> >**Figure 2 may be incorrect, as the Gaussian kernel graph should be fully connected, but the figure does not reflect this.**
>
> Thank you. We will update the Figure accordingly.

---

### Decision · Program_Chairs · 2025-05-01

**Decision:**

Accept (poster)

**Comment:**

The paper presents generalisation error bounds for graph neural networks (for both node-level and graph-level tasks) under the assumption that the graph is constructed on samples from a manifold. The bounds demonstrate the dependence of the generalisation error on the spectral continuity and the number of nodes. The work presents a new-dimension to the learning theory literature on GNNs by incorporating the manifold assumption, which could lead to further theoretical analysis for GNNs using the manifold connection. However, the immediate impact of the work could be limited due to lack of insight provided by generalisation bounds for NNs, and restriction of the work to convolutional GNNs.